# Influence on the Temperature Estimation by the Planetary Boundary Layer Scheme with Different Minimum Eddy Diffusivity in WRF v3.9.1.1

Hongyi Ding[1], Le Cao[1,*], Haimei Jiang[1], Wenxing Jia[1,3], Yong Chen[2], and Junling An[2]

[1]Key Laboratory for Aerosol-Cloud-Precipitation of China Meteorological Administration, Nanjing University of Information Science and Technology, Nanjing 210044, China
[2]State Key Laboratory of Atmospheric Boundary Layer Physics and Atmospheric Chemistry, Institute of Atmospheric Physics, Chinese Academy of Sciences, Beijing 100029, China
[3]Key Laboratory of Atmospheric Chemistry of CMA, Chinese Academy of Meteorological Sciences, Beijing 100081, China

**Correspondence:** L. Cao
(le.cao@nuist.edu.cn)

**Abstract.** The minimum eddy diffusivity (Kzmin) in the planetary boundary layer (PBL) scheme can influence the model performance in simulating meteorological parameters such as temperature. However, detailed studies on the sensitivities of the simulated temperature to the settings of Kzmin are still lacking. Thus, in this study, we evaluated the performance of the ACM2 (Asymmetrical Convective Model, version 2) scheme in the WRF (Weather Research and Forecasting) model with different settings of Kzmin, in simulating the spatiotemporal distribution of the temperature in the region of Beijing, China. Five constant values and a function were implemented in the model to calculate Kzmin, and the simulation results with different settings of Kzmin were compared and analyzed. The results show that the increase of Kzmin leads to an elevation of the 2-m temperature, especially in the nighttime. We figured out that the deviation in the 2-m temperature at night is mainly caused by the different estimations of the turbulent mixing under stable conditions in simulation scenarios with different Kzmin settings. Moreover, the spatial distribution of the temperature deviation indicates that under various underlying surface categories, the change in Kzmin exerts a distinct influence on the prediction of the 2-m temperature. This influence was found stronger during the nighttime than during the daytime, in plain areas than in mountain areas, in urban areas than in non-urban areas. In the nighttime of the urban areas, the influence on the simulated 2-m temperature brought about by the change in Kzmin is the strongest. In addition, the model performance using a functional type Kzmin in the ACM2 scheme in capturing the spatiotemporal distribution of the temperature in this region was also compared with that using a constant Kzmin.

## 1 Introduction

The planetary boundary layer (PBL) is a thin layer at the bottom of the atmosphere, which responds to a surface change within one hour or less (Stull, 1988). Generally, the height of the PBL is variable in time and space, ranging from hundreds of meters to a few kilometers. Moreover, within the PBL, a noticeable diurnal change in the temperature usually occurs, mainly caused by the warming and cooling of the ambient air by the ground surface during the daytime and the nighttime, through

the turbulent mixing. Turbulence in the PBL is an important form of air motion, and plays a critical role in vertically diffusing momentum, heat, moisture, and pollutants (Du et al., 2020). Therefore, it is essential to accurately estimate the effects of turbulence on the vertical mixing within the PBL in weather and air quality models.

In numerical models, the vertical mixing caused by turbulence is usually parameterized using PBL closure schemes. An appropriate PBL scheme can precisely capture the properties of the turbulent mixing as well as the structure of the PBL. At present, many PBL schemes are implemented in numerical models, such as YSU (Hong et al., 2006), MYJ (Janjić, 1994), MRF (Hong and Pan, 1996), ACM (Pleim and Chang, 1992), QNSE (Sukoriansky and Galperin, 2008; Sukoriansky et al., 2006), BouLac (Bougeault and Lacarrere, 1989), Shin-Hong scheme (Shin and Hong, 2015) , TEMF (Angevine, 2005; Angevine et al., 2010) and MYNN-EDMF (Olson et al., 2019). Generally, the PBL schemes can be classified into two types, local and non-local closure schemes. Local closure scheme, such as MYJ and BouLac, is also called K-theory (Stull, 1988). It usually determines the eddy diffusion coefficient from local prognostic variables such as the turbulent kinetic energy (TKE), local gradients of the wind speed and the potential temperature. However, in this type of PBL scheme, the mixing caused by large eddies is usually not adequately taken into account. As a result, the local closure schemes frequently fail in simulating the unstable boundary layer (Stull, 1988). In order to overcome the shortcomings of the local closure schemes, many non-local closure schemes such as MRF, YSU, Shin-Hong, TEMF, MYNN-EDMF and ACM have been proposed. In the MRF non-local closure scheme, a counter-gradient correction term is included (Hong and Pan, 1996), representing a contribution from the large-scale eddies to the total fluxes of heat, momentum and moisture. By comparing the model results with the observational data, Hong and Pan (1996) suggested that the MRF scheme simulates a more realistic structure of the daytime boundary layer than the local closure scheme. After that, based on MRF, a modified scheme, named YSU scheme was proposed (Hong et al., 2006), which treats the entrainment process occurring at the top of the PBL explicitly. It was found that the use of the YSU scheme tends to increase the boundary layer mixing in the thermally induced free convection regime, but tends to decrease the mixing in the mechanically induced forced convection regime (Hong et al., 2006). In 2015, Shin and Hong (2015) proposed a scale-aware scheme named Shin-Hong scheme. In this scheme, Shin and Hong (2015) introduced a new algorithm to estimate the vertical transport, so that the transport of the subgrid-scale heat is weakened. As a result, the predictions of large-eddy simulations (LES) can be better fitted. TEMF (Total Energy Mass-Flux) scheme, proposed by Angevine et al. (2010), is an update of EDMF (Eddy Diffusivity Mass-Flux) scheme (Angevine, 2005). In the TEMF scheme, the vertical mixing in free convective boundary layers is treated by combining eddy diffusivity and mass flux. In that way, it can estimate the non-local transport in the convective boundary layer more accurately and better represent the connection between dry thermals and cumulus clouds (Angevine et al., 2010). Recently, a non-local scheme named MYNN-EDMF was developed by Olson et al. (2019) by implementing an EDMF approach into the Mellor-Yamada-Nakanishi-Niino (MYNN) local scheme (Nakanishi and Niino, 2009). The EDMF approach adopted in MYNN-EDMF uses a mass-flux scheme to indicate the non-local turbulent mixing of heat, moisture, and momentum under convective conditions. Moreover, MYNN-EDMF defines TKE on mass points instead of at the interface of the grid cell, which makes the advection of TKE possible in this scheme (Olson et al., 2019). Aside from these non-local closure schemes, ACM (Asymmetrical Convective Model) scheme proposed by Pleim and Chang (1992) is a non-local PBL scheme that assumes that strongly buoyant plumes rise from the surface layer to all levels in the

convective boundary layer. It is also assumed in ACM that the downward motion between each adjacent layer is a gradual subsidence process. It was reported that the ACM scheme can improve the accuracy of the model in capturing the diffusion of chemicals released from elevated sources (Pleim and Chang, 1992). Based on that, by combining the original ACM with a local eddy diffusion module, Pleim (2007a,b) proposed the ACM2 scheme, to better represent both the super-grid and sub-grid components of the turbulent mixing in the convective boundary layer. They found that adding the local eddy diffusion module into the original ACM exerts a significant impact on quantities that have large surface fluxes, such as the momentum and the heat (Pleim, 2007a,b).

Many researchers have evaluated the performance of available PBL closure schemes under different meteorological conditions (Hu et al., 2010; Xie et al., 2012; Madala et al., 2014; Banks et al., 2016; Gunwani and Mohan, 2017). Generally, they found that during the PBL collapse and the nighttime, the PBL schemes are difficult to precisely capture the change of meteorological parameters such as the temperature. Moreover, they attributed the biases to three aspects: (1) Inaccurate calculation of the surface cooling rate. Chaouch et al. (2017) intercompared the performance of seven different PBL schemes in WRF (Weather Research and Forecasting) model (Skamarock et al., 2008) under foggy conditions in the United Arab Emirates. They found a cold bias in the 2-m air temperature during the PBL collapse and the nighttime, reflecting an overestimation of the surface cooling rate. Cuchiara et al. (2014) employed the WRF-Chem (WRF with Chemistry) model (Grell et al., 2005) to analyze the differences in the ozone prediction by four PBL schemes (YSU, ACM2, MYJ, QNSE). In their study, by comparing the model results with the observations, they found that the YSU scheme is in the best agreement with the observed ozone. Moreover, it was found by Cuchiara et al. (2014) that all these four PBL schemes predict a lower surface cooling rate, thus leading to an underestimation of the temperature by 2-3 K during the PBL collapse and the nighttime. (2) Unrealistic thermal coupling between the ambient air and the underlying surface in simulations. Udina et al. (2016) studied the vertical structure of a neutral and a stable PBL using the WRF-LES (WRF with Large Eddy Simulation) modeling system (Moeng et al., 2007). They suggested that in the model, the calculated thermal coupling at the surface is unrealistically large. As a result, the rate difference between the molecular thermal conduction and the vertical eddy diffusion is underestimated, leading to the prediction of a lower air temperature near the cooling surface in simulations. It also leads to the formation of a more stable boundary layer, compared to the observations. (3) Difference in internal properties of the PBL schemes. Shin and Hong (2011) numerically investigated the PBL properties using five PBL schemes (YSU, ACM2, MYJ, QNSE, BouLac) in WRF for a day in the Cooperative Atmosphere-Surface Exchange Study (CASES-99) field campaign (Poulos et al., 2002). They found that the simulated surface temperature and the 2-m temperature in the nighttime given by these five PBL schemes show positive biases, compared with the observations. In addition, they stated that the values of the minimum eddy diffusivity given in these PBL schemes are different, influencing the simulation results.

The minimum eddy diffusivity ($Kz_0$ or called Kzmin) is a small value to fix the estimation of the vertical eddy diffusivity (Kz) by the PBL closure schemes. It denotes a weak vertical diffusion in the free atmosphere or a strongly stable boundary layer that cannot be resolved by the model. Li and Rappenglueck (2018) investigated the causes behind the nighttime ozone biases in a simulation of the ground-level ozone in southeast Texas, US using the ACM2 scheme in CMAQ (Byun and Schere, 2006). They also compared the results using two different Kzmin settings. One is that the Kzmin is set as a constant value 1

$m^2 \, s^{-1}$ across the modeling domain, and the other setup is that Kzmin is computed by a formula so that it resides in a value range of 0.01-1.0 $m^2 \, s^{-1}$. They found that using the Kzmin calculated by the formula lowers the nighttime vertical mixing, and the average ozone bias is reduced compared with that using the alternative Kzmin setting. Their conclusions suggested that the setup of Kzmin is capable of changing the simulation results of the model. Nielsen-Gammon et al. (2010) evaluated the role of many parameters in the ACM2 scheme using WRF model. They found that the variation of Kzmin exerts a significant impact on the simulated temperature in the lower troposphere, especially at night. Moreover, Nielsen-Gammon et al. (2010) also suggested that in the upper troposphere, different values of Kzmin would cause a change in the intensity of the vertical mixing. As a result, different vertical profiles of the temperature and the water vapor were obtained in simulations using various Kzmin values, leading to a different prediction of cloud patterns and shortwave radiation.

However, to the present, detailed studies on the sensitivity of the temperature prediction to Kzmin are still lacking. Furthermore, the reasons causing the deviations in the simulated temperature brought by the change in Kzmin also need to be clarified. In addition, the effects of changing Kzmin on the temperature in areas with different categories of the underlying surface are also unclear. Thus, in this study, we performed a WRF model simulation on the meteorological field of the region near Beijing, China, and examined the impact exerted by the change in Kzmin on the simulated temperature. We also tried to figure out the mechanism for the change in the simulated temperature. By performing this numerical study, the role of Kzmin in the prediction of temperature in Beijing area of China can be clarified, which helps to determine the appropriate setup of Kzmin in temperature simulations across this region.

The structure of the paper is as follows. In Section 2, we describe the observational data, model settings and the PBL scheme used in the present study. In Section 3, simulation results and the related discussions are given. At last, major conclusions achieved in the present study are presented in Section 4. Future work is also prospected in this section.

## 2  Observational Data and Model Settings

In the present study, we first evaluated the performance of the PBL scheme (ACM2) with different Kzmin values in simulating observed meteorological parameters, and then assessed the connection between the differences in the temperature simulations and the value of Kzmin. At last, we used a function to calculate Kzmin and examined the performance of ACM2 with this function by comparing with that using a constant Kzmin (0.01 $m^2 \, s^{-1}$).

### 2.1  Observational Data

The observational data used in this study are provided by the Institute of Atmospheric Physics, Chinese Academy Sciences (IAP, CAS), obtained by a meteorological observation tower and an observational system of surface meteorological parameters. Aside from that, data provided by four automatic weather stations (AWS) (No. 54433, 54406, 54419, 54501) were also adopted to evaluate the model performance (see Fig. 1 for the locations of the IAP station and the AWS stations). The information of these observational data are as follows.

The meteorological observation tower of IAP was built in 1979, and consistently serves studies on air pollution, atmospheric boundary layer, and atmospheric turbulent diffusion. The tower is located at 39°58′N, 116°22′E and has a height of 325 m. A 15-level (8, 15, 32, 47, 63, 80, 102, 120, 140, 160, 180, 200, 240, 280, 320 meters) meteorological gradient observation system is instrumented on the tower, and provides data including wind speed, wind direction, temperature and moisture. The time resolution of the data is 10 minutes.

The observational system of surface meteorological parameters is instrumented next to the tower and provides the surface data including temperature, relative humidity, pressure, radiation, precipitation, wind speed and direction. The time resolution of these data is 30 minutes.

The observational data provided by AWS stations include wind speed, wind direction, temperature, moisture and surface pressure, with a time resolution of one hour. In the present study, we used the data of 2-m temperature and 10-m wind speed to evaluate the model performance. Among these AWS stations, No. 54433 station is located in the urban area of Beijing, similar to the IAP station. In contrast, the other three AWS stations (No. 54406, 54419, 54501) are located in rural or suburban areas of Beijing (see Fig. 1).

## 2.2   Model Description

In this study, we adopted the model WRF-ARW (Advanced Research WRF) version 3.9.1.1 to simulate the meteorological field of the region near Beijing, China. WRF is a mesoscale numerical weather forecasting system designed for atmospheric research and operational forecasting applications. The ARW version is developed and maintained by NCAR (National Center for Atmospheric Research) and is often used for scientific research. In the present study, the WRF model was adapted to the conditions of Beijing and its surrounding areas (Fig. 1). Three nested domains (D01, D02, and D03) were defined (see Fig. 1a), with horizontal grid spacings of 9 km (119×119 grid nodes), 3 km (196×193 grid nodes) and 1 km (259×259 grid nodes), respectively. Along the vertical direction, 48 levels were distributed. The terrain and the categories of the land use in the innermost domain (i.e., D03) are shown in Fig. 1(b) and (c). It is seen that there are mountains in the north (Yanshan Mountains) and the west (Taihang Mountains) of this area, and the North China Plain is located in the southeast of this studied domain. The boundary between the mountain area and the plain area is sharp. In the present study, two time periods (Jan. 8-15 and Jan. 20-24, 2014) were simulated. In these two time periods, the concentration of $PM_{2.5}$ (particulate matters with diameters smaller than 2.5 $\mu m$) accumulates (see Fig. S1 of the supplementary material), reflecting relatively stagnant weather conditions in this area. Moreover, these two selected time periods are mostly under sunny conditions, so that the complexity caused by the existence of clouds is minimized. The impact brought about by the presence of aerosols on the temperature is also not considered in the present study for simplicity. The simulation of each day starts at 8 LST (local standard time) of the day before the simulated day, due to the implementation of the spin-up process. The first 16 hours were treated as the spin-up time, and results obtained from the following 24-hour simulations were analyzed for the present study. Furthermore, the daytime and the nighttime in this study are defined as 8-17 LST and 18-7 LST, respectively. The initial and the boundary conditions were given by the 1°×1° National Centers for Environmental Prediction (NCEP) Global Forecast System (GFS) Final (FNL) gridded analysis datasets (National Centers for Environmental Prediction, National Weather Service, NOAA, U.S.

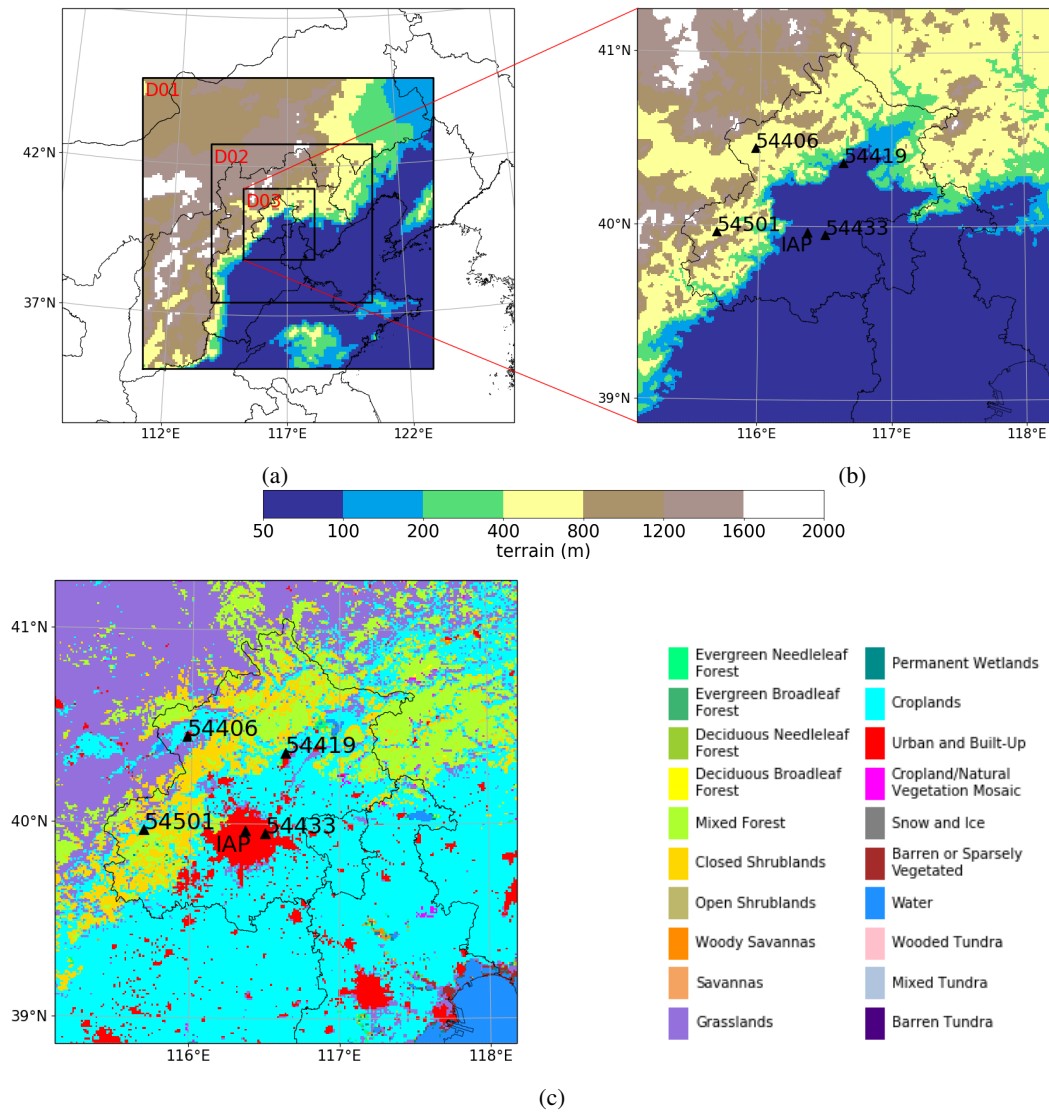

**Figure 1.** Description of (a) the locations of three nested domains, (b) an enlarged drawing of the terrain belonging to the innermost domain (i.e., D03), and (c) the spatial distribution of the land-use categories within D03. Locations of the IAP station and the four automatic weather stations (No. 54433, 54406, 54419, 54501) are also marked in (b) and (c).

Department of Commerce, 2000) and the Moderate Resolution Imaging Spectroradiometer (MODIS) dataset (Broxton et al., 2014) including 20 land-use categories. The parameterizations used in the present model are listed in Tab. 1.

**Table 1.** Parameterizations used in the present model.

| Namelist option | Description | Reference |
| --- | --- | --- |
| mp_physics | Purdue Lin scheme | Chen and Sun (2002) |
| ra_lw_physics | RRTM scheme | Mlawer et al. (1997) |
| ra_sw_physics | Dudhia scheme | Dudhia (1989) |
| sf_sfclay_physics | MM5 scheme | Zhang and Anthes (1982) |
| sf_surface_physics | Noah land surface model | Chen and Dudhia (2001) |
| bl_pbl_physics | ACM2 scheme | Pleim (2007a,b) |
| cu_physics | Grell 3D scheme (Domain 1 and 2) | Grell and Dévényi (2002) |
| sf_urban_physics | Single-layer UCM | Kusaka et al. (2001) |

### 2.2.1 ACM2 PBL Scheme

In this study, we adopted the ACM2 scheme as the PBL scheme. The reason for choosing ACM2 is that this scheme is included in many numerical models such as WRF (Skamarock et al., 2008) and CMAQ (Byun and Schere, 2006), and the settings of Kzmin in this scheme are given differently in these models, which will be described in a later context. The form of the scalar transport equation in ACM2 is as follows (Pleim, 2007a,b):

$$
\frac{\partial C_i}{\partial t} = f_{conv} Mu C_1 - f_{conv} Md_i C_i + f_{conv} Md_{i+1} C_{i+1} \frac{\Delta z_{i+1}}{\Delta z_i}
$$
$$
+ \frac{\partial}{\partial z}\left[ K_c \left(1 - f_{conv}\right) \frac{\partial C_i}{\partial z}\right] \tag{1}
$$

$$
f_{conv} = \frac{K_h \gamma_h}{K_c \gamma_h - K_h \dfrac{\partial \theta}{\partial z}} \tag{2}
$$

where $C_i$ is the predicted variable, such as the potential temperature in the $i$-th layer. $Mu$ is the mixing rate of the non-local upward convection, and $Md$ is the rate of the non-local downward mixing from the $i$-th layer to the $(i-1)$-th layer. $\Delta z_i$ is the thickness of the $i$-th model layer. $f_{conv}$ is a ratio factor weighting different contributions from non-local mixing and local mixing, and $\theta$ in Eq. (2) is the potential temperature. When the boundary layer is stable or neutral, the ACM2 scheme is mostly

**Table 2.** Scenarios simulated in the present study, with different setup of Kzmin in the ACM2 scheme.

| Type | Kzmin ($m^2 s^{-1}$) | Name |
|---|---|---|
| | 0.01 | ACM2_0.01 |
| | 0.2 | ACM2_0.2 |
| Constant | 0.5 | ACM2_0.5 |
| | 0.8 | ACM2_0.8 |
| | 1.0 | ACM2_1.0 |
| Function | 0.01~1.0 | ACM2_CMAQ |
| | 1.0 (daytime), 0.01 (nighttime) | AC_night_0.01 |
| Sensitivity test | | |
| | 1.0 (urban), 0.01 (non-urban) | AC_urban_1.0 |

dominated by the local transport process, which is represented by the last term on the right-hand side of Eq. (1). By adding the local transport term, the ACM2 scheme improves upon the ACM scheme in capturing the upward turbulent transport process within the boundary layer. (Pleim, 2007a,b).

### 2.2.2    Setup of Kzmin in ACM2

     In the ACM2 scheme instrumented in the WRF model, Kzmin is set as $0.01\,m^2\,s^{-1}$ by default. In contrast, in other numerical
models such as CMAQ (Byun and Schere, 2006), Kzmin is usually given a value between 0.001 and $1.0\,m^2\,s^{-1}$. Thus, in order to clarify the difference in simulation results caused by the variation of Kzmin, five simulation scenarios with different constant values of Kzmin were conducted in the present study (see Tab. 2). In addition, we also performed a simulation using a function to determine Kzmin (i.e., ACM2_CMAQ in Tab. 2). This function was taken from the CMAQ model, shown as follows:

$$Z \leq KZMAXL : \text{Kzmin} = 0.01 + (1 - 0.01)PURB \tag{3}$$

$$Z \geq KZMAXL : \text{Kzmin} = 0.01 \tag{4}$$

where:

$$KZMAXL = 500.0\,(\text{m}) \tag{5}$$

$$LU\_INDEX = LU\_INDEX(\text{Water}) : PURB = 0 \tag{6}$$

$$LU\_INDEX \neq LU\_INDEX(\text{Water}) :$$
$$PURB = \frac{Landusef(\text{Urban})}{1 - Landusef(\text{Water})} \tag{7}$$

In Eqs. (3)–(7), $Z$ is the height of the layer, and $KZMAXL$ is a prescribed height above which the atmosphere would not be significantly affected by the change in the surface properties. $PURB$ is a percentage ratio of the urbanization. $LU\_INDEX$ is an index representing the dominant category of the land use. $Landusef$ is a fraction of each land-use category in the grid cell. The spatial distributions of $Landusef$ as well as $PURB$ used in the present study are shown in Fig. S2 of the supplements. By

using the function described in Eqs. (3)–(7), the range of Kzmin given in the model is between 0.01 and 1 $\text{m}^2 \text{ s}^{-1}$. Moreover, for completely non-urban areas (i.e., $PURB = 0.0$), the value of Kzmin is 0.01 $\text{m}^2 \text{ s}^{-1}$, which is the same to the default value used in the ACM2 scheme of the WRF model, while for completely urban areas (i.e., $PURB = 1.0$), the value of Kzmin under the height of 500 m calculated by Eqs. (3)–(7) is 1.0, same to that used in the ACM2_1.0 scenario. We then compared the performance of ACM2 adopting this function with that using a constant Kzmin (0.01 $\text{m}^2 \text{ s}^{-1}$) in simulating the temperature in

the region of Beijing.

Furthermore, we designed two sensitivity tests in the present study (see Tab. 2). One of them is AC_night_0.01, in which Kzmin was set to 0.01 during the nighttime (same as ACM2_0.01), but 1.0 during the daytime (same as ACM2_1.0). The results of this sensitivity test help to differentiate the contributions from the difference in the simulated nighttime temperature and the change in Kzmin during the daytime. The other sensitivity test is AC_urban_1, in which Kzmin was set to 1.0 only

over urban areas (same as ACM2_1.0), but 0.01 over other areas (same as ACM2_0.01). Through this sensitivity test, the influence brought about by the temperature advection on the near-surface temperature estimation can be indicated, which will be discussed further in a later context.

### 2.3 Evaluation Criterion

In order to evaluate the performance of the model with different settings of Kzmin, four statistical metrics, index of agreement

(IOA) (Willmott, 1982), root mean square error (RMSE), correlation coefficient (R) and mean bias (MB) were implemented. These parameters are calculated as:

$$IOA = 1 - \left[ \frac{\sum\limits_{i=1}^{N} (P_i - O_i)^2}{\sum\limits_{i=1}^{N} \left( |P_i - \overline{O}| + |O_i - \overline{O}| \right)^2} \right] \tag{8}$$

$$\text{RMSE} = \sqrt{\frac{\sum\limits_{i=1}^{N} (P_i - O_i)^2}{N}} \tag{9}$$

$$\text{R} = \frac{\sum\limits_{i=1}^{N} \left(P_i - \overline{P}\right)\left(O_i - \overline{O}\right)}{\sqrt{\sum\limits_{i=1}^{N} \left(P_i - \overline{P}\right)^2}\sqrt{\sum\limits_{i=1}^{N} \left(O_i - \overline{O}\right)^2}} \tag{10}$$

$$\text{MB} = \frac{\sum\limits_{i=1}^{N} (P_i - O_i)}{N} \tag{11}$$

where $N$ is the number of data; $O$ is the observed value and $P$ is the value predicted by the model. $\overline{O}$ and $\overline{P}$ denote the average values of these variables. RMSE, R and MB are common statistical parameters and IOA is a metrics evaluating the fitness between model predictions and observations. When IOA is equal to 1, it represents a perfect match, while IOA=0 denotes that no agreement is achieved.

## 3   Results and Discussions

In Section 3.1, the performance of the model in simulating the 2-m temperature and the 10-m wind speed is evaluated and displayed. In Section 3.2, we show the impact of changing Kzmin on the 2-m temperature and discover the reasons for the change in the 2-m temperature. In Section 3.3, the effect of changing Kzmin under different underlying surface categories is shown. In Section 3.4, we compare the performance of ACM2 adopting the function described in Eqs. (3)–(7) with the results using the constant Kzmin 0.01 $\text{m}^2\,\text{s}^{-1}$.

### 3.1   Model Evaluations

The simulation results of the 2-m temperature and the 10-m wind speed were compared with the observational data provided by the above-mentioned IAP, CAS station and four automatic weather stations (AWS), to evaluate the model performance. The values of statistical parameters measuring the model performance are listed in Tab. 3. From a global view, the model behavior in capturing the 2-m temperature is satisfying. The correlation coefficients between the simulated temperature and the observations at these five stations reside in a value range of 0.78-0.94. Moreover, the index of agreement (i.e., IOA) also possesses a value above 0.75 for all these five stations. It was also found that the model performs better at the two urban stations (IAP and No. 54433) than at the other three rural stations, denoted by a smaller RMSE and a higher R (see Tab. 3). More information about the comparison between the simulated 2-m temperature and the observations at these five stations can be found in Sect. 3 of the supplementary material.

**Table 3.** Values of statistical parameters measuring the model performance in simulating the 2-m temperature (T2) and the 10-m wind speed (W10) at five observation stations.

| Station | T2 | | | | W10 | | | |
|---------|------|------|------|-------|------|------|------|------|
|         | RMSE | IOA  | R    | MB    | RMSE | IOA  | R    | MB   |
| IAP     | 2.79 | 0.84 | 0.94 | -2.49 | 3.21 | 0.26 | 0.64 | 2.51 |
| 54406   | 2.84 | 0.88 | 0.83 | 1.06  | 3.08 | 0.44 | 0.50 | 2.21 |
| 54419   | 3.16 | 0.85 | 0.86 | 1.11  | 2.28 | 0.30 | 0.35 | 1.49 |
| 54433   | 2.17 | 0.92 | 0.91 | -1.38 | 2.40 | 0.62 | 0.65 | 1.26 |
| 54501   | 4.85 | 0.76 | 0.78 | 2.75  | 2.06 | 0.52 | 0.36 | 0.94 |

Compared with the temperature estimation, the model predicts a higher wind speed at all these five stations (see MB of W10 in Tab. 3). The deviation between the simulation result and the observational data is more pronounced at the IAP station, as it possesses the largest MB of 2.51 m/s. Moreover, according to the correlation coefficient R, the simulated trend of the 10-m wind speed at two urban stations (IAP and No. 54433) is more consistent with the observations than that at the rural stations, as the correlation coefficient R at these two urban stations is above 0.6. More information about the simulated 10-m wind speed across the computational domain can also be found in Sect. 3 of the supplements.

Many factors can cause the deviation between the simulation results and the observational data, such as the uncertainties brought about by the imposed inaccurate initial and boundary conditions and the treatment of aerosols in the model as well as the choice of PBL schemes. However, the main objective of the present study is to estimate the influence caused by the change in Kzmin on the prediction of the temperature, rather than finding an improved PBL scheme that can more accurately reproduce the observations. Moreover, the change in Kzmin exerts a more significant influence on the temperature than other meteorological parameters such as the wind speed and the specific humidity (see Sect. 4 of the supplementary material). Thus, we paid more attention to the influence on the temperature prediction brought about by the change in Kzmin in the present study.

## 3.2 Impact of Changing Kzmin on 2-m Temperature

Figure 2 shows the diurnal mean time series of the temperature at 2 m (T2), the surface skin temperature (TSK), and the temperature at the first model layer (T_level1) at the observation site of IAP, predicted by ACM2 with different Kzmin constant values. In Fig. 2(a), it is seen that the highest T2 appears at approximately 15 LST (local standard time). At this time, the average T2 estimated by ACM2_0.01 and ACM2_1.0 are 1.81 °C and 2.25 °C, and T2 estimated by the other scenarios are between these two values. In contrast, the lowest T2 appears at about 8 LST. At this time, the average T2 predicted by ACM2_0.01 and ACM2_1.0 are -6.69 °C and -4.13 °C. Among these scenarios, ACM2_0.01 consistently predicts the lowest T2. Moreover, it was found that the simulated T2 elevates with the increase of Kzmin. In addition, the difference of T2 between these five

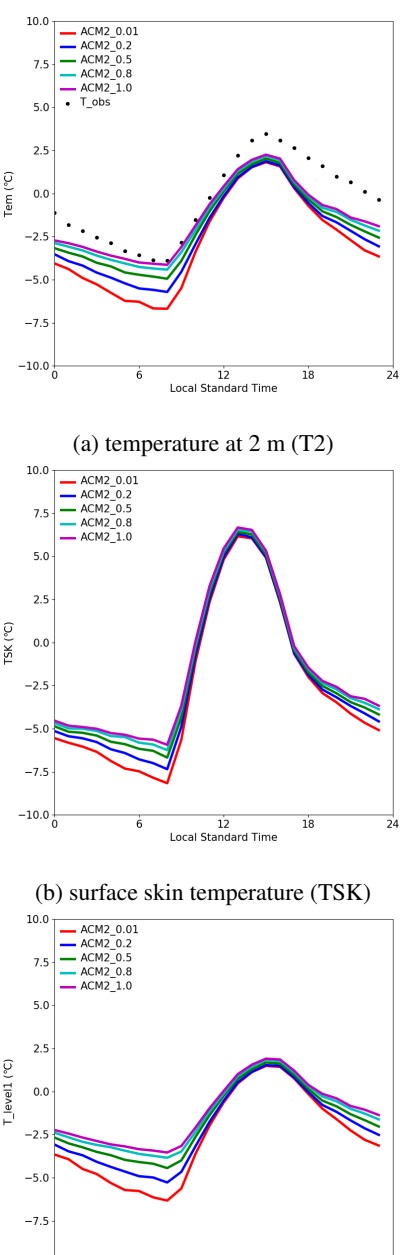

(a) temperature at 2 m (T2)

(b) surface skin temperature (TSK)

(c) temperature at the first model layer (T_level1)

**Figure 2.** Diurnal mean time series of (a) the temperature at 2 m (T2), (b) the surface skin temperature (TSK), and (c) the temperature at the first model layer (T_level1), predicted by the ACM2 scheme with different Kzmin constant values.

scenarios is smaller at a higher T2, while the difference becomes larger at a lower T2. As a result, the diurnal variation of T2 is reduced with the increase of Kzmin.

We then investigated the reasons causing the difference in the simulated T2 between these scenarios. In the model, T2 is calculated based on TSK, the surface sensible heat flux, and the exchange coefficient of temperature at 2 m. Moreover, the sensible heat flux is calculated according to the estimated TSK and the temperature at the first model layer (i.e., T_level1) (Li and Bou-Zeid, 2014). Thus, the estimation of T2 heavily depends on the values of the simulated TSK and T_level1. We thus show the diurnal mean time series of TSK and T_level1 estimated using different Kzmin values (see Fig. 2b and c). It can be

seen that during the nighttime, both TSK and T_level1 increase remarkably with the increase of Kzmin, which is similar to the temporal behavior of T2. This finding also partly agrees with the conclusions of Steeneveld et al. (2006), who stated that TSK increases substantially with an enhanced vertical mixing during the nighttime. However, from the temporal change in these two temperatures, we cannot figure out whether the difference in T2 is mostly caused by the change in the surface temperature (i.e., TSK) or the temperature in the atmosphere (i.e., T_level1), because of the interaction between the surface and the atmosphere.

Therefore, we continue to discover the dominant factor causing the change in TSK and T_level1, respectively.

We first try to infer the reason causing the difference in TSK, from the energy balance equation. In the Noah land surface model (Chen and Dudhia, 2001; Xie et al., 2012) used in this study, when neglecting the precipitation and the snow accumulated on the surface, the form of the energy balance equation is:

$$(1-\alpha)\,S\downarrow +L\downarrow -L\uparrow +G-HFX-LH = 0 \tag{12}$$

where $\alpha$ is the albedo of the underlying surface. $S\downarrow$ is the downward flux of the shortwave radiation. $L\downarrow$ is the downward flux of the longwave radiation emitted by the cloud and the atmosphere, and $L\uparrow$ is the upward flux of the longwave radiation emitted by the ground surface. $G$ is the ground heat flux, and it is positive when heat transfers from the soil to the surface. HFX is the sensible heat flux, and $LH$ is the latent heat flux at the surface. HFX and $LH$ are positive when the heat transfers from the surface to the atmosphere. We then combined $L\downarrow$ and $L\uparrow$ as a net longwave radiation flux ($NL = L\downarrow -L\uparrow$). As a result,

Eq. (12) becomes:

$$(1-\alpha)\,S\downarrow +G+NL-HFX-LH = 0 \tag{13}$$

Thus, five factors ($S\downarrow$, $G$, $NL$, HFX and $LH$) need to be evaluated for the difference of TSK between these simulation scenarios. Among these factors, we can first eliminate the shortwave radiation $S\downarrow$ as the dominant factor for the deviation in TSK. It is because that in this study, the difference in the downward shortwave radiation during the daytime between

280 scenarios using different Kzmin values is negligible (see Fig. S6 of the supplements). It means that the influences exerted by the shortwave radiation in the daytime under the conditions of various Kzmin settings are similar. Thus, it cannot result in the enlarged deviation in TSK during the nighttime through the carryover effects. Aside from that, the shortwave radiation at night is negligible. Therefore, we suggested that the shortwave radiation is unimportant for the deviation in the nighttime TSK prediction in the present study. Then four factors ($G$, $NL$, $LH$ and HFX) need to be evaluated. Figure 3 shows the temporal

profiles of the deviations (ACM2_0.2 minus ACM2_0.01, ACM2_0.5 minus ACM2_0.01, ACM2_0.8 minus ACM2_0.01,

ACM2_1.0 minus ACM2_0.01) in $G$, $NL$, $LH$ and HFX given by the model simulations. From Fig. 3(a), we can see that during the nighttime, the negative deviation in the ground heat flux $G$ becomes larger when Kzmin increases, denoting that $G$ is reduced with the increase of Kzmin in the nighttime. Because lower $G$ in the nighttime represents that less heat is transferred from the soil to the surface, which cannot lead to a higher TSK, the heat flux from the soil to the surface, $G$, can also be eliminated as the dominant factor causing the change in TSK during the nighttime. Then, from Fig. 3(b), it can be seen that during the nighttime, the negative deviation in the net longwave radiation (i.e., $NL$) becomes larger when Kzmin increases, which means that the value of $NL$ also gets reduced when Kzmin increases. Lower $NL$ means that the surface loses more longwave radiation energy, which cannot lead to a higher TSK. Thus, it can be deduced that the change in the net longwave radiation flux $NL$ is also not the major factor causing the growth of the TSK difference. Figure 3(c) shows that during the nighttime, there is no obvious difference in the latent heat flux $LH$ between these scenarios. Therefore, $LH$ can also be screened out. At last, Fig. 3(d) demonstrates that during the nighttime, the negative bias in the sensible heat flux HFX becomes larger when Kzmin increases, which means that HFX is reduced when Kzmin increases. Because lower HFX at night means that more heat is transferred from the atmosphere to the underlying surface, which can increase TSK. Thus, we can conclude that the difference in the sensible heat transported from the atmosphere to the ground among these simulation scenarios causes the different growth of TSK during the nighttime in the present simulations.

We then tried to reveal the reasons for the change in the air temperature at the first model layer (i.e., T_level1), caused by the modifications of Kzmin in the model. Figure 4 shows averaged vertical profiles of the potential temperature predicted by ACM2 using different Kzmin at 8 LST and 15 LST. From Fig. 4(a), we found that at 8 LST, the potential temperature difference at the first model layer is the largest between these five scenarios. When Kzmin increases, the predicted near-surface potential temperature elevates. It is consistent with the conclusion of Nielsen-Gammon et al. (2010) that Kzmin exerts the most prominent effect during the nighttime, and the variation of Kzmin is positively correlated with the change of the near-surface potential temperature. Moreover, seen from Fig. 4(a), the potential temperature difference becomes smaller at a higher altitude. Above the height of 400 m, the potential temperature profiles predicted by these five scenarios are almost identical. Therefore, when Kzmin increases, the vertical gradient of the mean potential temperature decreases at this time. It is because that during the nighttime, the PBL becomes stable, under which condition the turbulence is very weak. The settings of Kzmin thus exert a more significant influence on Kz. As a result, the increase of Kzmin would lead to a substantial enhancement of the vertical mixing during the nighttime. This enhanced vertical mixing then causes a more uniform vertical distribution of the potential temperature within the PBL and thus a prediction of a smaller temperature gradient below the top of the PBL. This conclusion also follows Nielsen-Gammon et al. (2010), who stated that the minimum vertical diffusivity is negatively correlated with the temperature gradient during the nighttime.

It should be noted that the difference in the longwave radiation emitted from the ground surface with various TSK is also a possible reason for the deviation in T_level1 between different scenarios. However, a comparison of temperature tendencies caused by the net longwave radiation at the first model layer between scenarios using different Kzmin values suggests that the net longwave radiation tends to reduce the nighttime temperature difference between these scenarios, instead of enlarging it

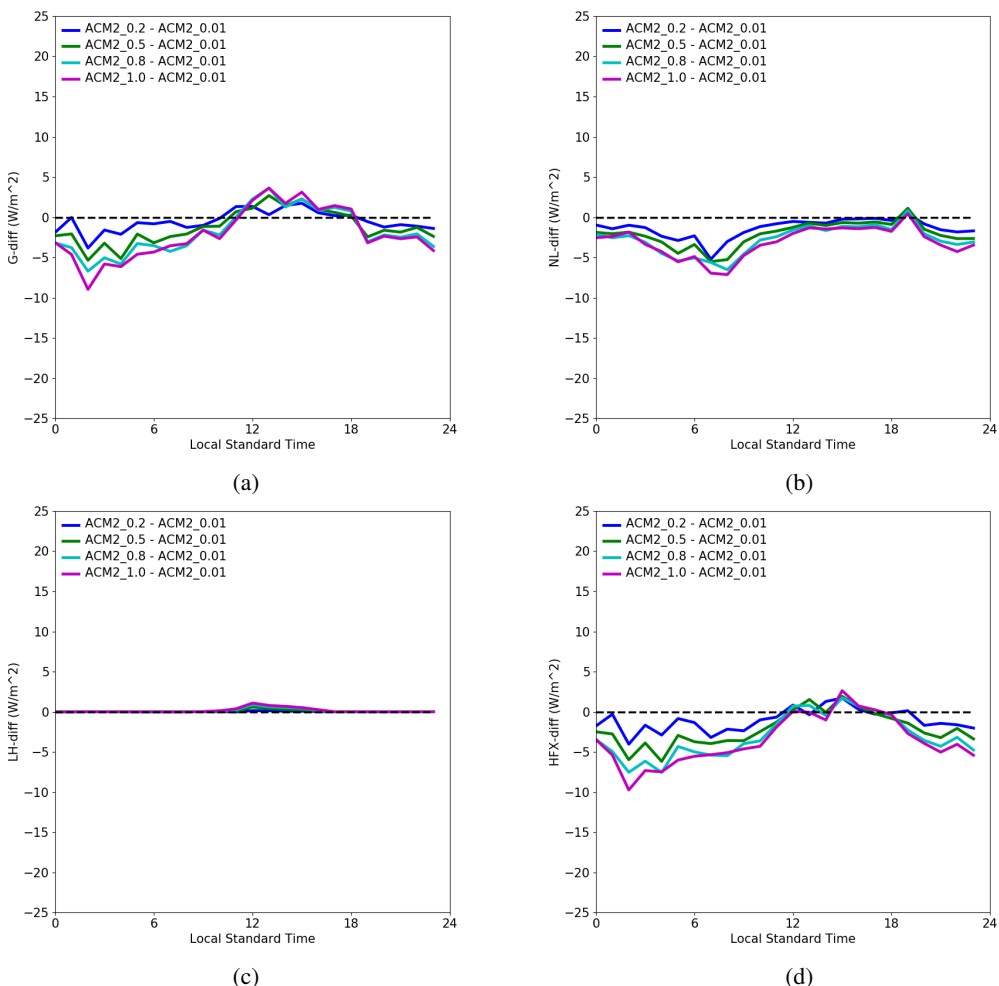

**Figure 3.** Diurnal mean time series of deviations (based on ACM2_0.01) in (a) the ground heat flux $G$, (b) the net longwave radiation flux $NL$, (c) the latent heat flux $LH$ and (d) the sensible heat flux HFX at the surface.

(see Sect. 4 of the supplementary material). Thus, the longwave radiation cannot be the factor causing the enlarged difference in the near-surface temperature between the nighttime simulations.

For the predicted vertical profile of the potential temperature at 15 LST, it was found in Fig. 4(b) that larger Kzmin also estimates a higher potential temperature. This deviation between the daytime temperature profiles can be partly attributed to the carryover effects of the significant temperature differences during the nighttime. Aside from that, in the ACM2 scheme,

Kzmin is added to Kz to constitute a total vertical turbulent diffusivity (Pleim, 2007a,b). As a result, even in the daytime when the turbulent diffusion is relatively strong, the change in Kzmin can still affect the turbulent mixing, resulting in a deviation in the predicted temperature in the daytime. The contributions of these two processes are to be investigated in a later context.

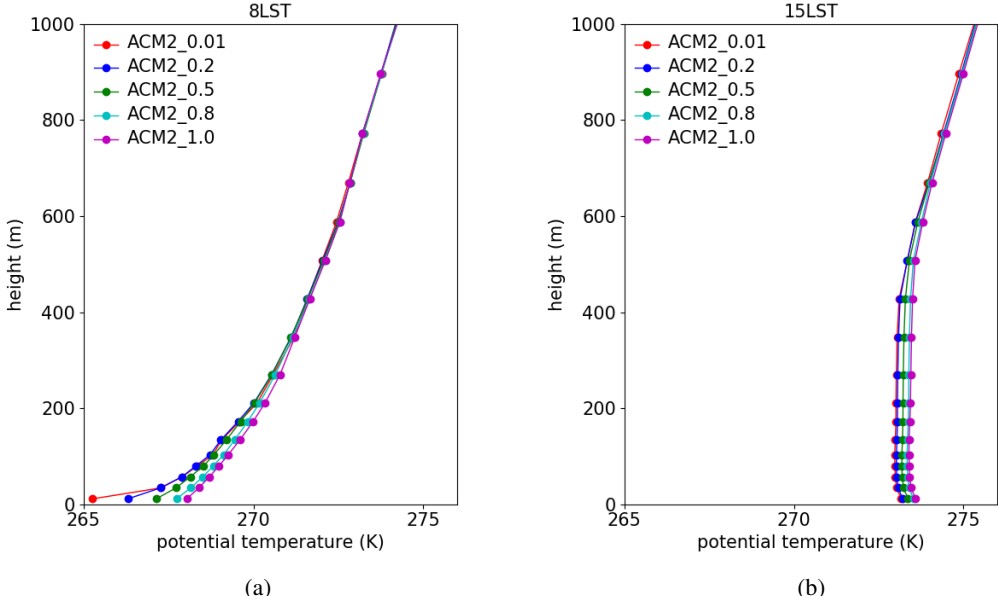

**Figure 4.** The vertical profiles of the potential temperature predicted by the ACM2 scheme with different Kzmin values at (a) 8 LST and (b) 15 LST, averaged over the simulated days.

Also, Fig. 4(b) shows that the temperature profiles at 15 LST are closer to each other than those at 8 LST. The reason is that the turbulent intensity is vigorous during the daytime, so that the change of Kzmin has a relatively minor impact on the eddy
diffusivity Kz and the vertical distribution of the temperature.

Based on the information given above, we can conclude that the differences in the simulated temperature during the daytime brought about by the change in Kzmin are caused by the combined effect of the large temperature difference during the nighttime and the different turbulent mixing intensity during the daytime. To clarify it, we designed another simulation scenario named AC_night_0.01, in which Kzmin was set to 0.01 during the nighttime (same as ACM2_0.01), but 1.0 during the daytime
(same as ACM2_1.0). By doing that, contributions to the difference of the temperature by these two processes can be assessed separately.

The time-averaged vertical profiles of the potential temperature at 8 LST and 15 LST are shown in Fig. 5. In Fig. 5(a), potential temperature profiles belonging to AC_night_0.01 and ACM2_0.01 were found close to each other, due to the same nighttime Kzmin values used in both scenarios. In contrast, in the daytime (see Fig. 5b), AC_night_0.01 was found predicting a
340 higher temperature than ACM2_0.01, which is caused by the increase of Kzmin during the daytime and the enhanced turbulent mixing. Meanwhile, AC_night_0.01 was also found giving a lower temperature than ACM2_1.0 during the daytime, although a same Kzmin (=1.0) is used during this time period in these two scenarios. Thus, the difference between AC_night_0.01 and ACM2_1.0 denotes the residual effect caused by the temperature difference during the nighttime.

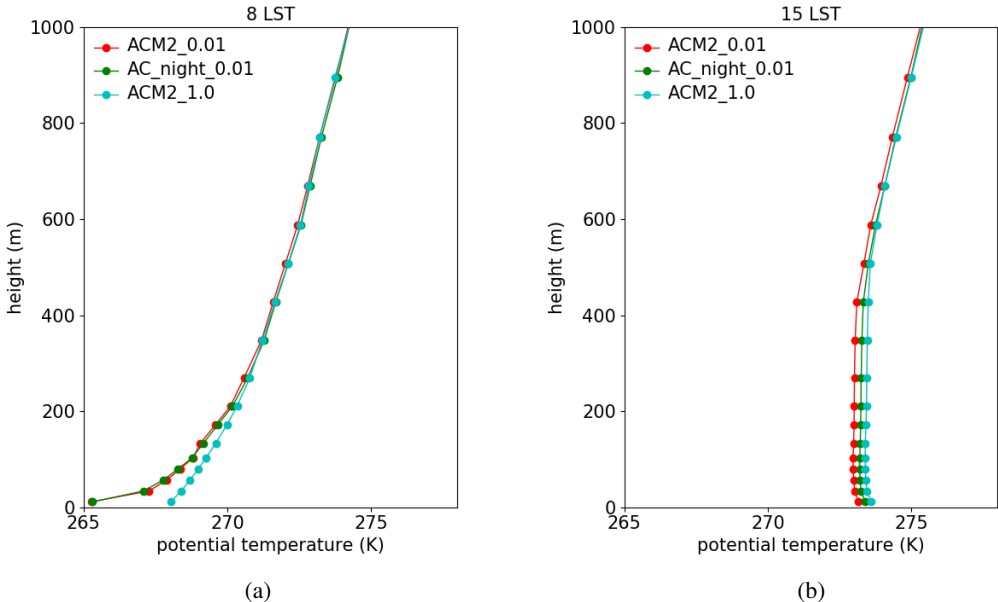

**Figure 5.** The vertical profiles of the potential temperature predicted by ACM2＿0.01, ACM2＿1.0 and AC＿night＿0.01 at (a) 8 LST and (b) 15 LST, averaged over the simulated days.

Thus, according to this sensitivity test, we confirmed that there are two primary processes causing the temperature difference during the daytime between scenarios using different Kzmin values. One is the residual effect caused by the change in Kzmin in the nighttime. It is because that different Kzmin results in a large deviation in the near-surface temperature during the nighttime. This deviation would maintain until the daytime comes so that the prediction of the daytime temperature would be affected. The other process is the change in Kzmin in the daytime. When Kzmin increases, the vertical mixing in the boundary layer is strengthened, which causes a stronger entrainment of the air from the upper layer into the boundary layer, thus resulting in a warmer boundary layer during the daytime.

This enhanced entrainment caused by the strengthening of the turbulent mixing during the daytime when ACM2 is used is also consistent with findings from previous studies. Unlike many other PBL schemes, ACM2 does not consider the entrainment flux explicitly. Instead, it includes the entrainment implicitly by combining a transilient term with the local mixing that is represented by the maximum of two forms of the turbulent diffusivity (Pleim, 2007a,b). Consequently, when ACM2 is used, the entrainment is very sensitive to the turbulent mixing within and above the PBL. It was also suggested by Nielsen-Gammon et al. (2010) and Hu et al. (2010) that when ACM2 is used, a stronger turbulent mixing in the boundary layer would result in a warmer PBL as well as a cooler free troposphere in the daytime. These conclusions confirm our suggestion in this study that a larger turbulent diffusivity given by ACM2 implementing a higher Kzmin leads to a strengthening of the entrainment and thus a warming of the boundary layer during the daytime.

Thus, based on the investigations of TSK and T_level1 discussed above, we can conclude the mechanism causing the remarkable change in T2 between the simulation scenarios with different Kzmin settings during the nighttime, shown in Fig. 2(a). When Kzmin increases, the vertical mixing in the nighttime is significantly enhanced. As a result, the near-surface temperature in the boundary layer (i.e., T_level1) is elevated due to the enhanced mixing of the warm air from the atmosphere above. The higher near-surface temperature thus leads to a reduction of the sensible heat flux at the surface (i.e., HFX) in the

nighttime and results in an increase of the surface skin temperature (TSK). Because the 2-m air temperature (i.e., T2) calculated in the model is positively dependent on the values of TSK and T_level1, the elevation of Kzmin thus causes the increase of T2.

### 3.3    Impact of Changing Kzmin under Different Underlying Surface Categories

The spatial distributions of the time averaged differences (ACM2_1.0 minus ACM2_0.01) in T2, TSK and HFX as well as the actual values obtained by ACM2_0.01 (the default Kzmin in WRF) over the daytime and the nighttime are shown in Fig. 6.

From Fig. 6(a) and (b), we can see three distinct features about the influence of increasing Kzmin on T2. First, the difference in T2 is mostly larger in plain areas than in mountain areas, which means that the increase of Kzmin has a stronger influence on T2 in plain areas than in mountain areas. The reason for the relatively stronger impact of changing Kzmin in plain areas than in mountain areas might be attributed to the spatial difference in the simulated near-surface wind speed throughout the computational domain. In Figs. S4 and S8 of the supplement, we displayed the spatial distributions of the 10-m wind speed

and the friction velocity during the nighttime, which is capable of representing the intensity of the wind shear. It was found that in mountain areas, the 10-m wind speed and the friction velocity are larger, compared with those in plain areas. It means that in mountain areas, a stronger wind shear is formed, which causes an enhancement of the turbulent mixing in the nocturnal boundary layer. Thus, a larger turbulent diffusivity were found in mountain areas rather than in plain areas (shown in Fig. S9 of the supplement).  As a result, elevating Kzmin in the mountain areas exerts a relatively minor influence on the vertical

mixing in the boundary layer as well as the simulated T2. Second, it was found that in plain areas, the difference of T2 is mostly larger during the nighttime than during the daytime, denoting a stronger impact on T2 exerted by the increase of Kzmin during the nighttime than during the daytime. Third, from the comparison between Fig. 6(a) and (b), it was found that during the nighttime, the difference of T2 is substantially larger in urban and built-up areas than that in areas with other land-use categories. But during the daytime, the difference is smaller. It means that the increase of Kzmin has the strongest influence on

T2 in urban and built-up areas during the nighttime. The reason why Kzmin plays a more important role in urban areas than in rural areas in the present study still needs further investigation. We guessed that it might be caused by the difference in physical properties (e.g., heat capacity) between areas with different land-use categories or the difference in parameterizations of some physical processes in the urban canopy model.

The different role of Kzmin in urban and rural areas is also able to modify the horizontal advection of temperature across the

computational domain, thus affecting the near-surface temperature prediction at the observation site. In order to clarify it, we designed another numerical experiment named AC_urban_1), in which Kzmin was set to 1.0 only over urban areas (same as ACM2_1.0), but 0.01 over other areas (same as ACM2_0.01). By doing that, the influence brought about by the temperature advection on the near-surface temperature estimation can be indicated.

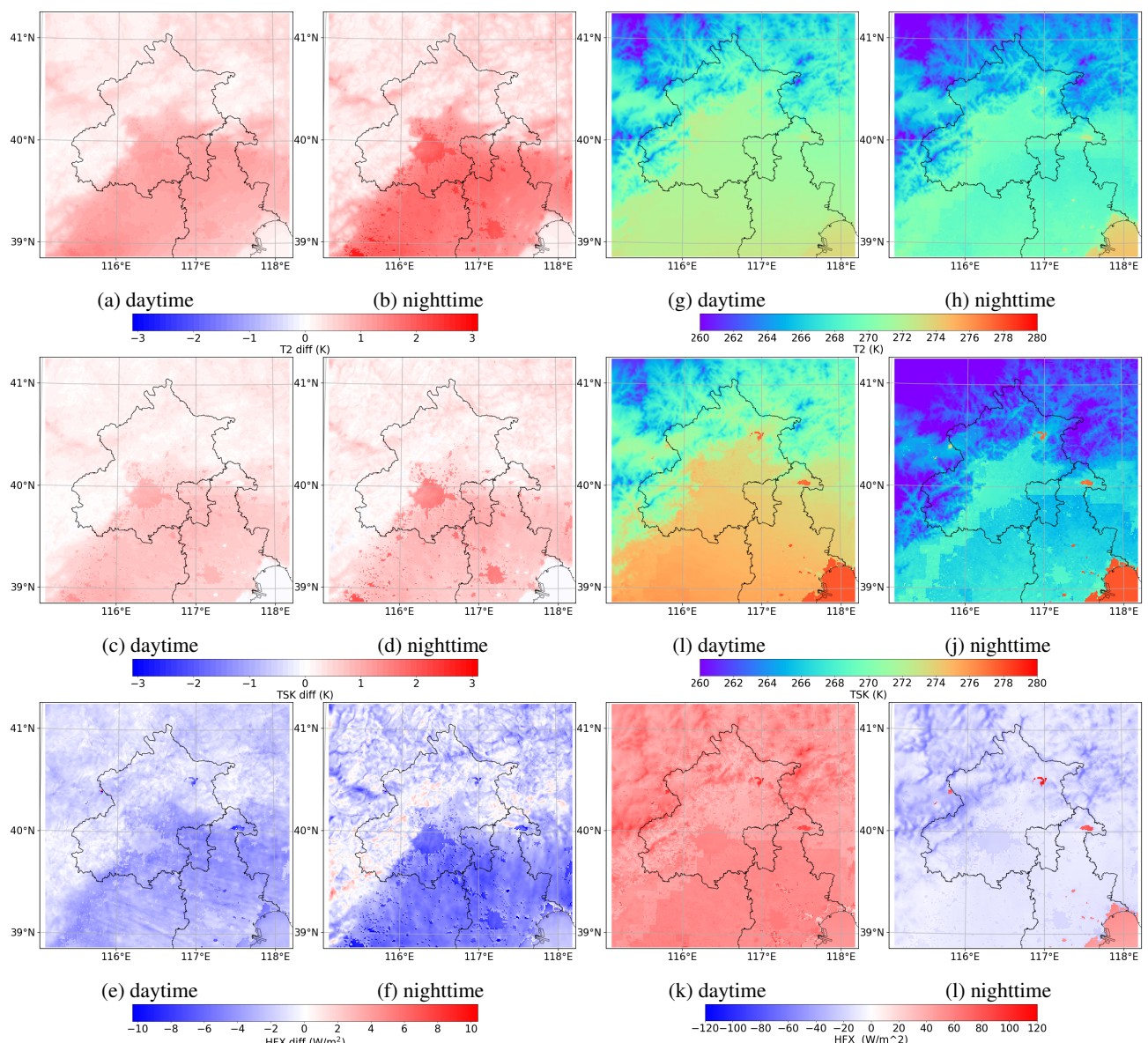

**Figure 6.** Spatial distribution of the mean difference (ACM2_1.0 minus ACM2_0.01) in (a, b) the 2-m temperature (T2), (c, d) the surface skin temperature (TSK), and (e, f) the sensible heat flux (HFX) over the daytime and the nighttime. The actual values of (g, h) T2, (i, j) TSK, and (k, l) HFX simulated by ACM2 with the default Kzmin value (i.e., ACM2_0.01) during the daytime and the nighttime are also displayed for reference.

The time-averaged vertical profiles of the potential temperature at the observation site (i.e., IAP station) at 8 LST and 15
LST are shown in Fig. 7. It was found that although AC_urban_1 and ACM2_1.0 possess a same value of Kzmin for the

urban areas that are focused on in this study, AC_urban_1 still estimates a lower nighttime temperature than ACM2_1.0 (see Fig. 7a), due to the smaller Kzmin over rural areas. We suggested the reason as that lower Kzmin over rural areas in AC_urban_1 causes a weaker turbulent mixing and thus a lower near-surface temperature in rural areas than those given by ACM2_1.0. This difference in the near-surface temperature of rural areas consequently affects the temperature prediction over urban areas through the advection process. In contrast, the nighttime temperature difference between AC_urban_1 and ACM2_0.01 shown in Fig. 7(a) can be mostly attributed to the stronger turbulent mixing over urban areas in AC_urban_1 relative to that in ACM2_0.01. Because of that, the vertical gradient of the near-surface temperature is reduced in AC_urban_1. ACM2_urban_1 thus predicts a higher temperature than ACM2_0.01 near the surface. Therefore, we can conclude that the difference in the near-surface temperature at the observation site at 8 LST between scenarios using different Kzmin values can be attributed to the combined effect of the change in the local Kzmin and the altering of Kzmin in other areas through the advection process. This conclusion also holds for the simulated near-surface temperature at 15 LST, shown in Fig. 7(b).

Regarding TSK, in Fig. 6(c) and (d), we can see that the three features obtained in the analysis of T2 are also valid. The difference of TSK is larger in plain areas than in mountain areas, during the nighttime than during the daytime, in urban areas than in areas with other land-use categories at night. But the difference of TSK is less than that of T2, indicating that the increase of Kzmin exerts a less influence on TSK than on T2.

From Fig. 6(e) and (f) we can see that in most areas, when Kzmin increases, HFX decreases during both the daytime and the nighttime, which represents that less heat is transferred from the surface to the atmosphere or larger amount of heat is transported from the atmosphere to the surface, respectively. Moreover, it was shown that the difference of HFX is larger in plain areas than in mountain areas. In addition, by comparing Fig. 6(e) and (f), we found the boundary of the urban areas clearly discernible during the nighttime but unclear during the daytime. It denotes that in the nighttime, the effects of changing Kzmin on HFX in urban and non-urban areas are substantially different, while in the daytime, the effects are similar.

## 3.4   Performance of ACM2_CMAQ and ACM2_0.01

We then adopted a function described in Eqs. (3)–(7) to calculate Kzmin and compared the performance of the model (i.e., ACM2_CMAQ) with that using a constant Kzmin (ACM2_0.01). The diurnal mean time series of T2, TSK, HFX predicted by ACM2_CMAQ and ACM2_0.01 are shown in Fig. 8. From Fig. 8(a), we can see that T2 predicted by ACM2_CMAQ is consistently higher than that predicted by ACM2_0.01, although it still underestimates the observations. It is also shown that the difference in T2 between ACM2_CMAQ and ACM2_0.01 increases at a lower T2, and the difference attains the greatest when T2 reaches the lowest value in the morning. The difference in the minimum T2 between ACM2_CMAQ and ACM2_0.01 is 2.01 °C, while the deviation in the maximum T2 between these two scenarios is only 0.17 °C. As a result, the diurnal change of T2 using ACM2_CMAQ is smaller than that using ACM2_0.01. From Fig. 8(b), it is shown that the behavior of the predicted TSK belonging to these two scenarios is similar to that of T2. ACM2_CMAQ predicts a higher TSK than ACM2_0.01 especially at night and thus a smaller diurnal change of TSK. From Fig. 8(c), we found that the difference of HFX between ACM2_CMAQ and ACM2_0.01 is larger at a negative HFX, while the difference is negligible at a positive HFX. When HFX is negative, the value of HFX predicted by ACM2_CMAQ is lower than that predicted by ACM2_0.01.

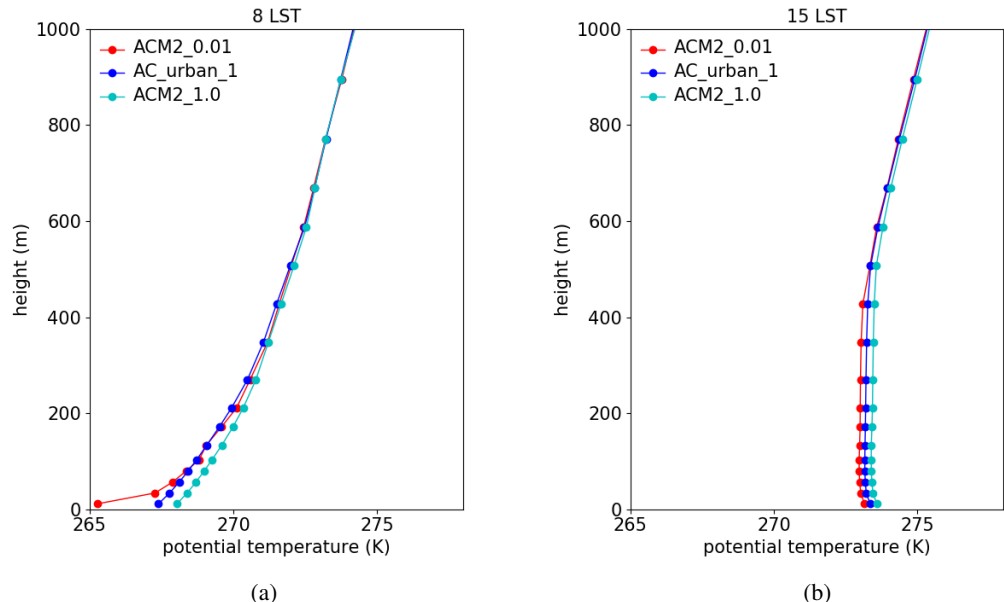

**Figure 7.** The vertical profiles of the potential temperature predicted by ACM2_0.01, AC_urban_1, ACM2_1.0 at (a) 8 LST and (b) 15 LST, averaged over the simulated days.

**Table 4.** Statistical performances of ACM2_0.01 and ACM2_CMAQ in simulating T2.

| Scenarios | RMSE | IOA | R | MB |
|-----------|------|-----|-----|------|
| ACM2_0.01 | 2.79 | 0.84 | 0.94 | -2.49 |
| ACM2_CMAQ | 1.95 | 0.90 | 0.91 | -1.44 |

It means that in the night simulations using ACM2_CMAQ, more heat is transferred from the atmosphere to the ground, relative to the ACM2_0.01 scenario. It is caused by the enhanced vertical mixing within the boundary layer during the night-time in the ACM2_CMAQ scenario, as the Kzmin value given in ACM2_CMAQ is higher than that in ACM2_0.01. Table 4 summarizes the statistical performances of these two scenarios in simulating T2. It was found that the correlation coefficient (R) of ACM2_0.01 is higher than that of ACM2_CMAQ, which denotes that ACM2_0.01 predicts a better trend of the

change in T2 than ACM2_CMAQ. However, the index of agreement (i.e., IOA) of ACM2_CMAQ is closer to 1.0 than that of ACM2_0.01, and RMSE of ACM2_CMAQ is smaller than that of ACM2_0.01, denoting that the magnitude of T2 simulated by ACM2_CMAQ deviates less from the observation than that by ACM2_0.01.

  Figure 9 shows the averaged vertical profiles of the potential temperature at 8 LST and 15 LST predicted by ACM2_0.01 and ACM2_CMAQ as well as the observations. It can be seen that at 8 LST, the difference of the potential temperature

between these two simulation scenarios is remarkable near the ground. Below the height of 100 m, the potential temperature

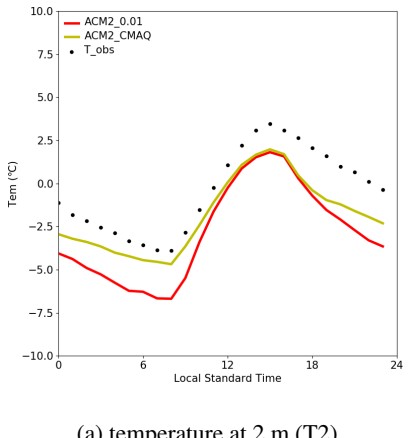

(a) temperature at 2 m (T2)

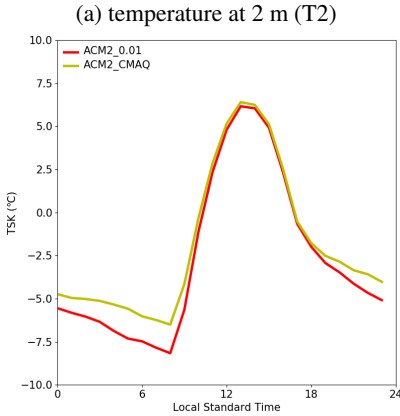

(b) surface skin temperature (TSK)

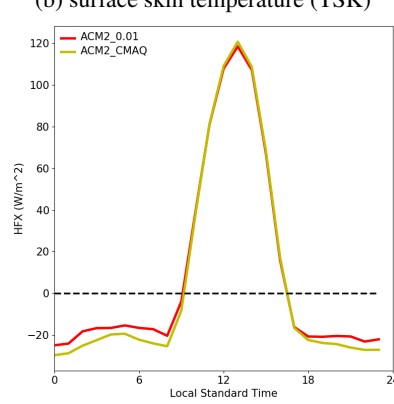

(c) sensible heat flux at the ground (HFX)

**Figure 8.** Diurnal mean time series of (a) T2, (b) TSK, and (c) HFX, predicted by ACM2_CMAQ and ACM2_0.01.

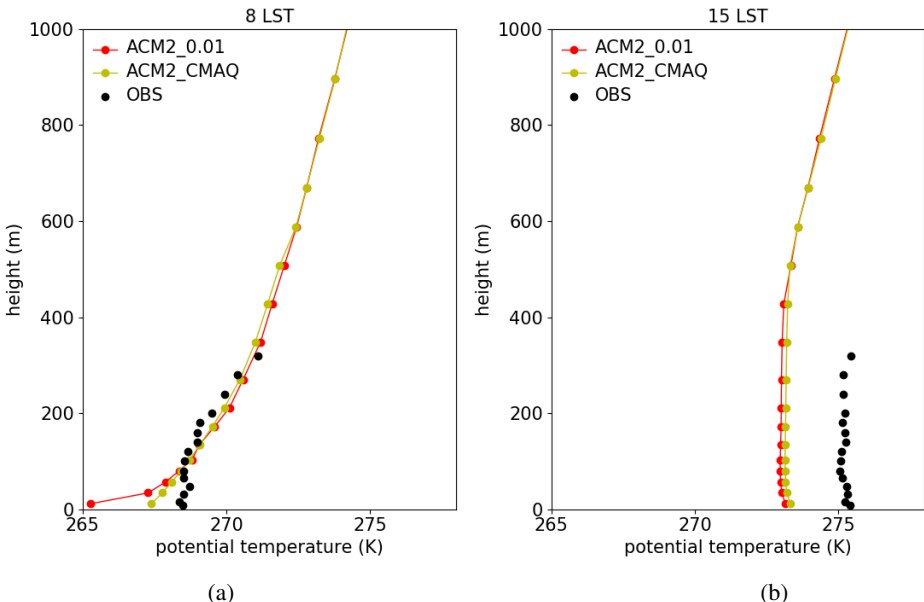

**Figure 9.** The vertical profiles of the potential temperature predicted by ACM2_0.01 and ACM2_CMAQ as well as the observations at (a) 8 LST and (b) 15 LST, averaged over the simulated days.

estimated by ACM2_CMAQ is higher than that estimated by ACM2_0.01, and the largest difference (more than 2 K) occurs in the lowest layer of the model. In contrast, between the heights of 100 m and 500 m, the potential temperature predicted by ACM2_CMAQ is slightly lower than ACM2_0.01. The potential temperature gradient estimated by ACM2_CMAQ is thus smaller than that estimated by ACM2_0.01 below the height of 500 m. This different prediction of the vertical gradient of the potential temperature is because that at 8 LST, the turbulent mixing is very weak so that Kzmin dominates Kz. Moreover, the value of Kzmin calculated by the function in ACM2_CMAQ is larger than that of ACM2_0.01 below the height of 500 m. Therefore, ACM2_CMAQ estimates a stronger vertical mixing, thus reducing the potential temperature gradient below the height of 500 m. In contrast, the profiles of the potential temperature predicted by these two scenarios are similar above the height of 500 m, because of the equal Kzmin values above 500 m in these two scenarios. By comparing the simulation results with the observations (see Fig. 9a), we found that ACM2_CMAQ estimates a closer potential temperature profile to the observations compared with ACM2_0.01, but it still overestimates the temperature gradient in the boundary layer at this time.

At 15 LST (see Fig. 9b), both the potential temperatures predicted by these two scenarios are about 2 K lower than the obervations. The potential temperature predicted by ACM2_CMAQ is slightly higher than that predicted by ACM2_0.01 below the height of 500 m. Above 500 m, there is only a minor difference between ACM2_0.01 and ACM2_CMAQ. The reason is the same to that in the 8 LST simulation that above 500 m, the values of Kzmin given in ACM2_0.01 and ACM2_CMAQ are equal.

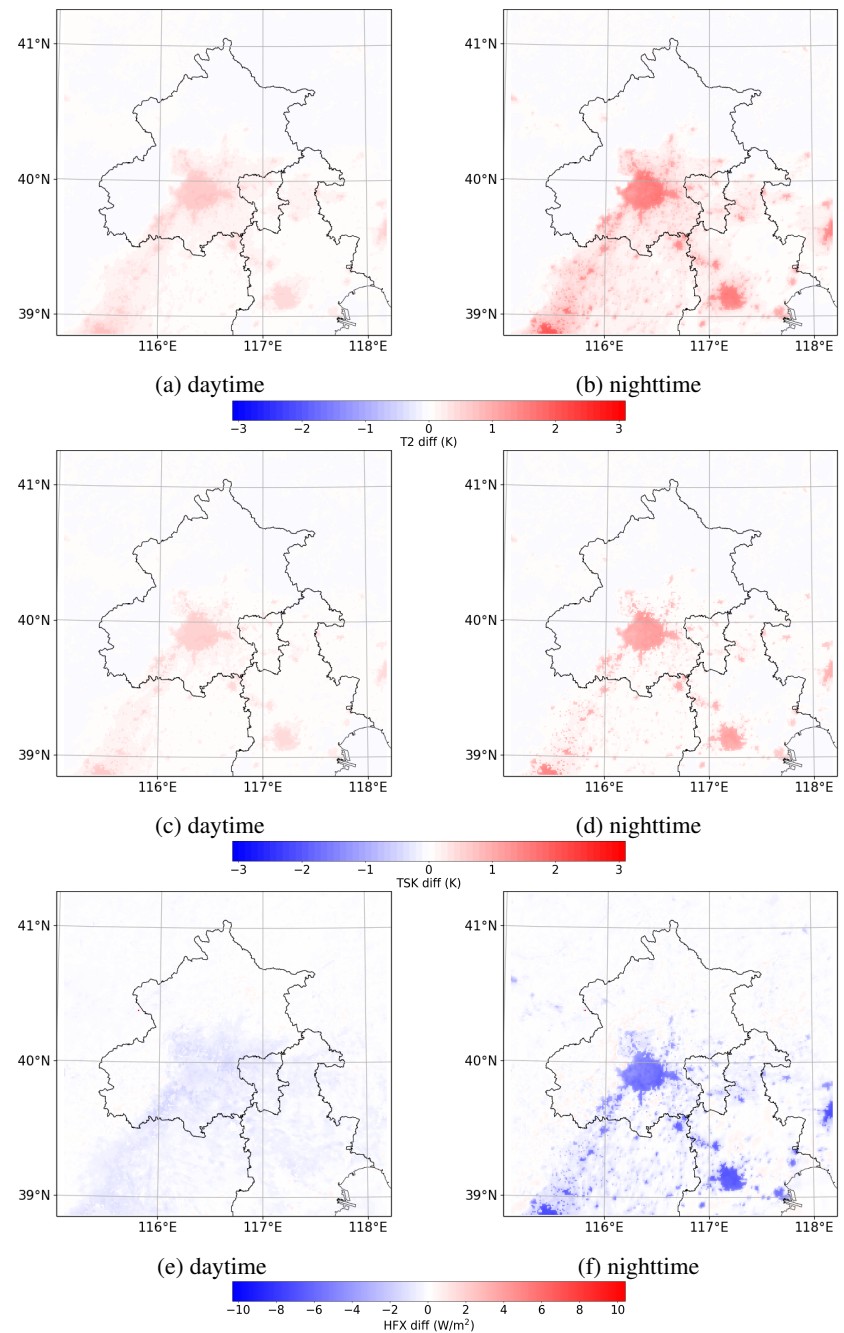

**Figure 10.** Spatial distribution of the mean differences (ACM2_CMAQ minus ACM2_0.01) in (a, b) T2, (c, d) TSK, and (e, f) HFX over the daytime (left row) and the nighttime (right row).

The spatial distribution of the time averaged differences of T2, TSK and HFX between ACM2_CMAQ and ACM2_0.01 is shown in Fig. 10. From Fig. 10(a)-(d), we can see that in urban areas, the difference in T2 and TSK between these two scenarios is mostly positive during both the daytime and the nighttime, which denotes that T2 and TSK predicted by ACM2_CMAQ are consistently higher than those predicted by ACM2_0.01 in urban areas. But in non-urban areas, the difference is minor. It is because that compared with ACM2_0.01, ACM2_CMAQ uses a larger Kzmin in urban and built-up areas below the height of 500 m. As a result, ACM2_CMAQ estimates a stronger vertical mixing in the PBL of urban areas than ACM2_0.01, thus resulting in an elevation of T2 and TSK. In contrast, in non-urban areas, the Kzmin values given in these two scenarios are identical (i.e., 0.01), the simulation results are thus similar. By comparing Fig. 10(a) and (b), it can also be found that in urban areas, the difference in T2 between ACM2_0.01 and ACM2_CMAQ during the nighttime is larger than that during the daytime, and this feature is also valid for the TSK deviation, according to Fig. 10(c) and (d). These results are consistent with the conclusions achieved above, stating that the change in Kzmin has the largest impact on the variation of T2 and TSK in the nighttime of the urban areas. With respect to HFX, it was found in Fig. 10(e) that during the daytime, the difference in HFX between ACM2_CMAQ and ACM2_0.01 is indiscernible. But in the nighttime (see Fig. 10f), the shapes of the urban areas are clearly indicated in the spatial distribution of the HFX deviation. It means that the HFX difference in urban areas between ACM2_0.01 and ACM2_CMAQ mostly exists during the nighttime. In addition, Fig. 10(f) also shows that during the nighttime, the negative value of HFX in urban areas predicted by ACM2_CMAQ is lower than that provided by ACM2_0.01, which means that larger amount of heat is transferred from the atmosphere to the ground during the nighttime in the ACM2_CMAQ simulation.

## 4  Conclusions and Future Developments

In this study, we evaluated the performance of the ACM2 scheme with different Kzmin settings in the estimation of the 2-m temperature (T2), the surface skin temperature (TSK) and the near-surface air temperature (T_level1) in the area of Beijing, China. We found that the change in Kzmin in the ACM2 scheme can significantly influence the model performance in simulating these temperatures. Increasing Kzmin leads to a remarkable elevation of T2 at night and a weakening of the diurnal change of T2. From the energy balance equation, we figured out that the mechanism for the elevation of T2 at night is because larger Kzmin causes a significant enhancement of the turbulent mixing within the stable boundary layer at night. Then the enhanced mixing in the nighttime reduces the vertical gradient of the potential temperature within the boundary layer, and thus elevates the air temperature near the ground surface (i.e., T_level1). The elevation of the near-surface air temperature then decreases the night sensible heat flux at the ground (i.e., HFX in the model), representing that larger heat is transferred from the atmosphere to the ground. As a result, the surface temperature (i.e., TSK) becomes higher. The elevations of TSK and T_level1 in the model consequently lead to the increase of the 2-m temperature (T2).

We also figured out the features about the influence of changing Kzmin on the temperature prediction under different underlying surface categories. It was found that the impact on the 2-m temperature and the surface temperature brought by the

change in Kzmin is stronger during the nighttime than during the daytime, in plain areas than in mountain areas, in urban areas than in non-urban areas at night.

When using a function calculating Kzmin in the ACM2 scheme (i.e., the ACM2_CMAQ scenario), we found that the simulated 2-m temperature elevates in urban areas, compared with that using a constant Kzmin (i.e., ACM2_0.01). The reason is the same as that in the nighttime simulation, larger Kzmin in ACM2_CMAQ leads to a transport of more sensible heat from the atmosphere to the surface, resulting in a higher prediction of the 2-m temperature. In addition, the simulated vertical profiles of the potential temperature show that ACM2_CMAQ estimates a smaller potential temperature gradient than ACM2_0.01 within the boundary layer, especially at night, and the profile of the potential temperature given by ACM2_CMAQ is closer to the observation than that provided by ACM2_0.01. Moreover, the spatial distribution of the temperature deviation between these two scenarios shows that in the daytime, the temperature simulated by ACM2_CMAQ is only slightly higher than ACM2_0.01 in both urban and non-urban areas. But the difference becomes remarkable in the nighttime of the urban areas.

The present study has some limitations. For instance, currently we are lack of the observational data representing the surface energy balance and surface exchange fluxes, as these data may help to better evaluate the model performance. Moreover, in the present study, the influence of changing Kzmin on the temperature prediction was investigated based on the ACM2 scheme. The role of Kzmin in other PBL schemes such as YSU (Hong et al., 2006) and QNSE (Sukoriansky and Galperin, 2008; Sukoriansky et al., 2006) should also be studied in the future. In addition, the conclusions achieved in the present study are primarily valid for the area of Beijing, China. Thus, whether these conclusions are still valid in other areas especially those with different categories of the underlying surface (i.e., sea, desert) also needs to be clarified.

In the future, more extended time periods are to be simulated so that the conclusions achieved in the present study can be verified more thoroughly. Moreover, the impacts of changing Kzmin on the spatiotemporal distribution of other meteorological parameters such the wind and moisture will also be evaluated more thoroughly. In addition, we plan to assess the effects of changing Kzmin on simulations of air pollution under different weather conditions, due to the strong connection between the diffusion of pollutants and the vertical turbulent mixing.

*Code and data availability.* The source code of WRF version 3.9.1.1 can be found on the website: www2.mmm.ucar.edu/wrf/users/download/ (Skamarock et al., 2008). The code described by Eqs. (3)–(7), defining a functional type Kzmin in the ACM2 scheme of WRF, can be found in the directory named "Modified_WRF_Code" in the supplementary material of the present manuscript. The WRF model input namelist file and the post-processing python scripts are also available in the supplements, named "WRF_namelist" and "post-processing-scripts", respectively. In addition, the observational data obtained from the meteorological observation tower as well as the observational system, provided by the Institute of Atmospheric Physics, Chinese Academy Sciences (IAP, CAS), are included in the directory "obs_data" of the supplements.

*Acknowledgements.* This work was financially supported by the National Key R&D Program of China (Grant No. 2017YFC0209801) and the National Natural Science Foundation of China (Grant No. 41705103). The authors also like to thank Dr. Holger Grosshans from

Physikalisch-Technische Bundesanstalt (PTB) for revising this manuscript. The numerical calculations in this paper have been done on the high performance computing system in the High Performance Computing Center, Nanjing University of Information Science & Technology.

*Author contributions.* Hongyi Ding and Le Cao conceived the idea of the article and ran the model. Hongyi Ding also wrote the python script for the data processing. Le Cao and Hongyi Ding wrote the paper together. Haimei Jiang and Wenxing Jia revised the paper and gave valuable suggestions. Yong Chen and Junling An from IAP, CAS provided the observational data and gave useful advice on the comparison of the model results with the observations. All the authors listed have read and approved the final manuscript.

*Competing interests.* The authors declare no conflict of interest.

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
