# Peer review of "Influence on the Temperature Estimation by the Planetary Boundary Layer Scheme with Different Minimum Eddy Diffusivity in WRF v3.9.1.1"

_Geoscientific Model Development, 2021_

## Author Response (AR1)

**Response to Referee #1**

The authors sincerely thank Reviewer #1 for the valuable comments and the very helpful considerations, which greatly contribute to an improvement of our paper.

In the following, we address the particular issues raised by Reviewer #1:

**Q1.1:** One of my major concerns is the lack of a proper model evaluation. The evaluation of the modelling setup is currently spread throughout different sections and is mainly focused on 2m temperature from one observational point (section 2.1), and a few vertical temperature profile (Fig 7). I would highly recommend to perform an extensive evaluation for 2m temperature and 10m-wind speed at different observational stations across the domain. This could offer insight on the bias in near surface temperature and atmospheric stability during the night. Ideally an evaluation of the modelled surface energy balance and surface exchange fluxes should be provided for the observational point where the physical process analysis is conducted (in section 3.1). Such an evaluation would help the reader to understand how the model performs at the default Kzmin setting, before the effects of the changes in Kzmin are investigated.

**A1.1:** Thanks a lot for the suggestion. According to the comments of the reviewer, we performed a comparison of other meteorological parameters such as the 10-m wind speed with the observational data. Moreover, we adopted observations from other four AWS (Automatic Weather Stations) for the evaluation of the model. Among these AWS stations, the No. 54433 AWS station is located in the urban area of Beijing, similar to the IAP station described in the present study, while the other three AWS stations (No. 54406, 54419, 54501) are located in rural or suburban areas of Beijing.

The values of statistical parameters measuring the model performance are listed in Tab. A1 of this rebuttal. From a global view, the model behavior in capturing the 2-m temperature is satisfying. The correlation coefficients between the simulated temperature and the observations at these five stations reside in a value range of 0.78-0.95. Moreover, the index of agreement (i.e. IOA) also possesses a value above 0.75 for all these five stations. It was also found that the model performs better at the two urban stations (IAP and No. 54433) than at the other three rural stations, denoted by a relatively smaller RMSE and a higher R (see Tab. A1).

In contrast to that, the model tends to predict a higher wind speed at all these five stations (see MB of W10 in Tab. A1). The deviation between the simulation result and the observational data is more pronounced at the IAP station, as it possesses the largest MB of 2.51 m/s. Moreover, from the values of the correlation coefficient R, it was found that the simulated trend of the 10-m wind speed at two urban stations (IAP and No. 54433) is more consistent with the observations than that at the rural stations, as the correlation coefficient R at these two urban stations reaches a value above 0.6.

Table A1. Statistical parameters for simulations of the 2-m temperature and the 10-m wind speed at five observation stations

| Station | T2 | | | | W10 | | | |
|---------|------|------|------|-------|------|------|------|------|
| | RMSE | IOA | R | MB | RMSE | IOA | R | MB |
| IAP | 2.79 | 0.84 | 0.94 | -2.49 | 3.21 | 0.26 | 0.64 | 2.51 |
| 54406 | 2.84 | 0.88 | 0.83 | 1.06 | 3.08 | 0.44 | 0.50 | 2.21 |
| 54419 | 3.16 | 0.85 | 0.86 | 1.11 | 2.28 | 0.30 | 0.35 | 1.49 |
| 54433 | 2.17 | 0.92 | 0.91 | -1.38 | 2.40 | 0.62 | 0.65 | 1.26 |
| 54501 | 4.85 | 0.76 | 0.78 | 2.75 | 2.06 | 0.52 | 0.36 | 0.94 |

Figure A1 of this rebuttal also shows the diurnal change of the simulated T2 at these five stations as well as the observations. It can be seen that at these stations, the highest T2 appears at approximately 15:00 local standard time (LST), while the lowest T2 appears at around 8:00 LST. This is also the reason why we focused on these two time points in the present study. Moreover, it was also found that at the three rural stations (No. 54406, 54419 and 54501), the simulated 2-m temperature at these stations is higher than the observational value in the nighttime, while it is lower than the observation during the daytime when the temperature is high. As a result, the simulated diurnal variation of T2 is weaker than the observations at these rural stations. On the contrary, at two urban stations, the cold bias appears during the whole day.

[Figure]

Figure A1. Diurnal change of the time-averaged 2-m temperature (T2) at five observation stations.

With respect to the 10-m wind speed, Fig. A2 shows the spatial distribution of the time-averaged 10-m wind speed over the nighttime. It was found that a larger wind speed is estimated in mountain areas rather than in plain areas, leading to a stronger wind shear in mountain areas compared with that in plain areas, which will be discussed further in a later response to the question **Q1.4**.

Evaluations of the 2-m temperature and the 10-m wind speed at these five observation stations are added in the revised manuscript as a section "3.1 Model Evaluations". Please see lines 211-237 in the revised manuscript.

[Figure]

Figure A2. The spatial distribution of the time-averaged 10-m wind speed over the nighttime.

Unfortunately, currently we are lack of observational data for the surface energy balance and surface exchange fluxes, so that we are unable to perform this comparison at present. We considered it as one of the limitations of the present study and added the related sentences in the last section, please see lines 502-503 of the revised manuscript.

**Q1.2:** The physical process analysis in section 3.1 lacks depth. The authors claim that shortwave radiation is unimportant for the deviations in Tskin during nighttime, because shortwave radiation in negligible during night. Still the amount of shortwave incoming radiation during the day affects the daytime evolution of air temperature and atmospheric stability in the boundary layer, which can impact the nocturnal temperature gradient and the atmospheric stability during the night. This might result in different deviations for the Tskin due to changes in Kzmin under sunny and cloudy conditions. Have the authors investigated the effect of Kzmin during different meteorological conditions? I would recommend that the authors discuss in more detail potential daytime carryover effects on the nocturnal near surface temperature differences.

**A1.2:** In this study, all the two winter periods we investigated are mostly under sunny conditions. Under this condition, the difference in the downward shortwave radiation between scenarios using different Kzmin values during the daytime is negligible (see Fig. A3 of this rebuttal). It means that the influences exerted by the shortwave radiation in the daytime under the conditions of various Kzmin settings are similar, which screens out the possibility that the enlarged deviation of TSK during the nighttime is caused by different longwave radiations in the daytime through the carryover effects. Thus, we suggested that the shortwave radiation is unimportant for

the deviation in the nighttime temperature prediction in the present study. Additional explanations are added in the revised manuscript. Please see lines 273-280.

[Figure]

Figure A3. Diurnal change of the downward shortwave radiation at the ground surface.

The reason we chose the sunny days for the investigation of the present study is to simplify the problem, as the situation with the existence of clouds is very complex. The change of Kzmin would affect the spatiotemporal distribution of the temperature and the moisture, which results in a different estimation of the clouds. Then the different estimation of the clouds would influence the radiation and thus the temperature. In addition, many other potential influences would also be induced. Therefore, for simplicity, we only adopted sunny conditions in this study. We provided the related information in lines 144-145 of the revised manuscript.

**Q1.3:** Have the authors investigate the importance of temperature advection on the changes in near surface temperature gradience and the sensible heat flux? Although it makes sense that changes in Kzmin affect HFX, the authors do not report if advection of temperature is present over the observational site and whether it contributes to changes in near surface temperature gradience. Considering that figure S.2 shows substantial spatial variability in Kz it is reasonable to assume that there will be spatial differences in near surface temperature, which could lead to substantial temperature advection. Thus, I would recommend that the authors either discuss the importance of temperature advection or show that it is negligible at the location of the analysis.
**A1.3:** Thanks a lot for pointing out the importance of the horizontal advection. In order to figure out the role of the horizontal advection in the present study, we designed another numerical experiment as follows.

In this numerical experiment (namely AC_urban_1), Kzmin was set to 1.0 only over urban areas (same as ACM2_1.0), but 0.01 over other areas (same as ACM2_0.01). By doing that, the influence brought about by the temperature advection can be

indicated.

The time-averaged vertical profiles of the potential temperature at the observation site (i.e. IAP station) at 8 LST and 15 LST are shown in Fig. A4 of this rebuttal. It was found that although AC_urban_1 and ACM2_1.0 possess a same value of Kzmin for the urban areas that are focused on in this study, AC_urban_1 still estimates a lower temperature than ACM2_1.0 (see Fig. A4(a)), due to the smaller Kzmin over rural areas. We suggested the reason as that lower Kzmin over rural areas in AC_urban_1 causes a lower simulated temperature in rural areas than that given by ACM2_1.0. This difference in the temperature of rural areas consequently affects the near-surface temperature over urban areas through the advection process. In contrast, the temperature difference between AC_urban_1 and ACM2_0.01 in Fig. A4(a) can be mostly attributed to the stronger turbulent mixing over urban areas in AC_urban_1 relative to that in ACM2_0.01. As a result, the vertical gradient of the near-surface temperature is reduced in AC_urban_1. ACM2_urban_1 thus predicts a higher temperature than ACM2_0.01 near the surface.

Therefore, we can conclude that the difference in the near-surface temperature at the observational site at 8 LST between scenarios using different Kzmin values is attributed to the combined effect of the change in the local Kzmin and the altering of Kzmin in other areas through the advection process.

[Figure]

Figure A4. The vertical profiles of the potential temperature predicted by ACM2_0.01, ACM2_urban_1, ACM2_1.0 at (a) 8 LST and (b) 15 LST, averaged over the simulated days.

We have added the related sentences and the discussions in lines 387-404 of the revised manuscript. Thanks again for the valuable suggestion.

**Q1.4:** The analysis on the spatial differences in 2m temperature and the impact of

Kzmin (section 3.2) is brief and not well substantiated. Why do area with complex terrain produce larger eddies? Are there spatial differences in near surface wind speed throughout the domain that could explain the differences in Kz? The near surface wind speed during night-time could be very important for the nocturnal turbulent mixing, but is not discussed at all in the manuscript. Why is the T2m over urban areas strongly affected by Kzmin? I would expect that over urban areas there is more turbulent mixing during night-time and therefore the Kzmin values would be less relevant, as their effect is mainly dominant during very stable atmospheric condition. Did the authors use an urban canopy model to parameterize physical processes of the urban surface? If not (seem to be the case based on Table 1), what value can the Kzmin analysis provide over urban areas if the physical processes responsible for the surface energy balance and turbulent exchange fluxes over the cities are not properly parameterized?

**A1.4:** Thanks a lot for the comment. The reviewer is correct. The difference in the intensity of the wind shear actually determines the role of Kzmin in areas with different land-use categories. In Fig. A5 of this rebuttal, we displayed the spatial distribution of the friction velocity during the nighttime, which is capable of representing the intensity of the wind shear. It was found that in mountain areas, the friction velocity is larger, compared with that in plain areas. It means that in the mountain areas, a stronger wind shear is formed, which then causes a stronger turbulent mixing and a larger Kz than those in plain areas. As a result, the influence of changing Kzmin is more pronounced in mountain areas than in plain areas. We have rephrased the expressions in the revised manuscript. Please see lines 368-377.

[Figure]

Figure A5. The spatial distribution of the friction velocity during the nighttime.

Regarding to the urban canopy model, in the present study, we used the single-layer urban canopy model (UCM), we have added this information in Tab. 1 of the revised manuscript. However, we are still unclear about why Kzmin plays a more important

role in urban areas than in rural areas. We guessed that the stronger influence over urban areas than over non-urban areas by the change of Kzmin might be caused by the difference in some physical properties (e.g. heat capacity) between areas with different land-use categories or the difference in parameterizations of some physical processes in the UCM. We have added the related discussions in lines 383-386 of the revised manuscript.

**Q1.5:** A discussion on the limitations of the current research approach and a comparison with results from previous studies, on the effects of eddy diffusivity in the nocturnal boundary layer, is missing.

**A1.5:** Thanks. We added more discussions about the limitations of the present study in the conclusion section (i.e. Section 4) of the revised manuscript. Please see lines 502-508 of the revised manuscript. Moreover, we added more contents comparing our results and conclusions with those obtained in previous studies. Please see lines 347-355.

**Specific comments:**

**Q1.6:** Section 2.2 Have the authors allowed for any model spin-up time to ensure that the atmospheric state and soil temperatures are properly a spun-up before the analysis is conducted?

**A1.6:** Yes, we do implemented a spin-up process. We actually computed 40 hours for the simulation of each day, starting at 8 LST of the day before the simulated day. The first 16 hours were then treated as the spin-up time, and results obtained from the following 24-hour simulations were used for the analysis of the present study. We added the related description in lines 146-148 of the revised manuscript.

**Q1.7:** Equation 7. Here PURB seem to be dependent only on the urban and water fractions. What happens when the landuse fraction is neither urban nor water (e.g., vegetated areas)?

**A1.7:**   The reviewer is correct saying that the calculation of PURB only depends on the fractions of urban and water belonging to this grid cell. In the model, each grid cell possesses values of many land-use category fractions (e.g., urban, water, croplands), and the values of these fractions are between 0 and 1 for each grid cell. Moreover, the index lu_index denotes the dominant category of this grid cell. For example, in vegetated areas (e.g. dominated by plant) asked by the reviewer, the plant fraction is the largest among all the fractions of this grid cell. The urban and the water fractions also have values but should be smaller than the plant fraction. Then, by using the values of urban and water fractions, we were able to calculate PURB for this grid cell.

The spatial distribution of PURB used in the present study is shown in Fig. A6 as follows. This figure as well as the related explanations are also added in the revised

supplement of the manuscript, see .

[Figure]

Figure A6. The spatial distribution of PURB across the computational domain.

**Q1.8:** Line 225. Please note that differences in the surface outgoing longwave radiation, could affect the cooling/heating rates of temperature at the different model levels. The WRF model does allow for the output of temperature tendencies due to changes in net-shortwave and net-longwave (at all model levels). It might be worth utilizing these tendency terms to identify if there are indirect effects on the near surface temperature due to the differences in net longwave radiation between the experiments.

**A1.8:** Thanks for the advice. As we know, higher TSK causes the release of a stronger longwave radiation from the surface, which is able to increase the temperature at different model layers. Moreover, a higher temperature at a specific model layer could also result in a loss of a larger amount of energy through the emission of longwave radiation. We thus output the temperature tendencies caused by the change in net longwave radiation at all model layers under the condition of different Kzmin values, and we found the difference in the temperature tendencies more pronounced near the surface, which means that the change caused by the net longwave radiation under different Kzmin conditions is more obvious at the first layer of the model.

We then plotted the diurnal change of the mean potential temperature and the hourly temperature tendencies due to the net longwave radiation at the first model layer, given by ACM2_0.01 and ACM2_1.0 (see Fig. A7), to assess the contribution of the net longwave radiation to the difference in the near-surface temperature between scenarios using different Kzmin values. In Fig. A7(b), a negative temperature tendency was found in these two scenarios during the nighttime. It means that at the first model layer, the net longwave radiation leads to a loss of the net energy at night. Moreover, Fig. A7(b) shows that the temperature tendency of ACM2_1.0 is lower

than that of ACM2_0.01 during the nighttime. It denotes that the decrease of temperature caused by the net longwave radiation in ACM2_1.0 is more rapid than that in ACM2_0.01. As a result, the influence of the net longwave radiation is to reduce the temperature difference between these two scenarios, especially during the nighttime, as ACM2_1.0 estimates a higher first-level temperature (i.e. theta_level1) at night than ACM2_0.01 (see Fig. A7(a)). Thus, the net longwave radiation is not the factor causing the enlarged temperature difference between the nighttime simulations of ACM2_1.0 and ACM2_0.01.

We have extended the related discussions in lines 312-317 of the revised manuscript.

[Figure]

Figure A7 Diurnal changes of the mean potential temperature and the temperature tendency due to changes in net longwave radiation at the first model layer.

**Q1.9:** Line 248. The turbulence within the boundary layer cannot be resolved (only parameterized) with the current WRF setup as the horizontal grid spacing is too large. Kzmin is more important during nighttime, because under very stable conditions the modelled diffusivity (Kz) is lower than Kzmin, in which case the boundary layer scheme will replace Kz with Kzmin is the calculation of exchange coefficients and heat fluxes.

**A1.9:** Yes, the statement here about the reason why Kzmin is more important under stable conditions is inappropriate in the original manuscript. We rephrased the sentences in the revised manuscript. Please see line 306.

**Q1.10:** Line 255. The increase in entrainment is very much depended on how the entrainment is parameterized in the ACM2 scheme, information which the authors do not provide. For instance, If the modelled entrainment is proportional to the surface sensible heat flux (this is common in some PBL schemes), then is barely any change in entrainment as Fig. 3d shows minimal change in sensible heat flux during daytime. In any case, it would be important that the authors describe how they concluded that entrainment increases for higher Kzmin and how they calculated it.

**A1.10:** Thanks for the comment. Unlike many other PBL schemes, ACM2 does not consider the entrainment flux explicitly. Instead, it includes the entrainment implicitly by combining a transilient term with the local mixing that is represented by the maximum of two forms of the eddy diffusivity Kz (Pleim 2007a, 2007b). It was found that when ACM2 is used, the entrainment is very sensitive to the mixing within the PBL, and it was also suggested by Nielsen-Gammon et al. (2010) and Hu et al. (2010) that when ACM2 is used, a stronger eddy mixing in the boundary layer would result in a warmer PBL as well as a cooler free troposphere. Thus, we suggested that a larger Kz given by ACM2 implementing a higher Kzmin leads to the strengthening of the entrainment and thus a warming of the boundary layer during the daytime.

We have refined the related description in the manuscript, and added more explanations. Please see lines 347-355 in the revised manuscript.

**Q1.11:** Line 259 Is the difference between the daytime temperature profiles caused only by effects of Kzmin on the turbulence during daytime or is it purely due to the already large temperature differences during night-time?

**A1.11:** The reviewer is correct saying that the nighttime bias can accumulate so that it would impact the daytime simulation. However, in ACM2, the implementation of Kzmin can be simply represented as follows:

$$Kz'=Kz+Kzmin,$$

which means that Kzmin is added to Kz to constitute a total vertical turbulent diffusivity Kz'. As a result, even though in the daytime when Kz is relatively large, Kzmin is still able to affect Kz' in areas that the turbulent mixing is relatively weak. We have added this information as well as the explanations in lines 319-323 of the revised manuscript.

Based on the information given above, we can conclude that the difference in the simulated temperature during the daytime brought about by the change of Kzmin is caused by the combined effect of the large temperature difference during the nighttime and the different turbulent mixing intensity during the daytime. To clarify it, we performed another numerical experiment. In this experiment (named AC_night_0.01), Kzmin was set to 0.01 during the nighttime (same as ACM2_0.01), but 1.0 during the daytime (same as ACM2_1.0). By doing that, the contributions to the difference of the temperature by these two processes can be assessed separately.

The time-averaged vertical profiles of the potential temperature at 8 LST and 15 LST are shown in Fig. A8. In Fig. A8(a), potential temperature profiles belonging to AC_night_0.01 and ACM2_0.01 were found close to each other, due to the same values of the nighttime Kzmin used in these two scenarios. In contrast to that, in the daytime (see Fig. A8(b)), AC_night_0.01 was found predicting a higher temperature than ACM2_0.01, which is caused by the increase of Kzmin during the daytime and the enhanced turbulent mixing.

In contrast, AC_night_0.01 was also found giving a lower temperature than ACM2_1.0, although a same Kzmin (=1.0) is used during the daytime in these two scenarios. Thus, the difference between AC_night_0.01 and ACM2_1.0 denotes the residual effect caused by the bias of the temperature during the nighttime.

[Figure]

Figure A8. The vertical profiles of the potential temperature predicted by ACM2_0.01, ACM2_1.0 and AC_night_0.01 at (a) 8 LST and (b) 15 LST, averaged over the simulated days.

Thus, according to this numerical experiment, we confirm that there are two primary processes causing the temperature difference during the daytime. One is the change of Kzmin during the daytime. When Kzmin is increased, the vertical mixing in the boundary layer is enhanced, which causes a stronger transport of the air from the upper layer into the boundary layer, resulting in a warmer boundary layer during the daytime. The other process is the residual effect caused by the change of Kzmin in the nighttime. It is because that different Kzmin results in a large deviation in the near-surface temperature during the nighttime. This deviation would maintain until the daytime comes so that the prediction of the daytime temperature would also be affected.

The corresponding results and the related discussions are added in the revised manuscript. Please see lines 327-346.

**Q1.12:** Line 265. The 2m temperature in WRF is not interpolated based on Tskin and the temperature at the first model level, but is rather calculated from the Tskin temperature, the surface sensible heat flux and the exchange coefficient of temperature at 2m (as seen in Li and Bou-Zeid, 2014).

**A1.12:** Thanks a lot for the comment. Our statement here about the interpolation is inappropriate. In the model, T2 is calculated based on the TSK, the surface sensible heat flux and the exchange coefficient of temperature at 2m. Moreover, the sensible heat flux is calculated according to the estimated TSK and the temperature at the first atmospheric level (T_level1) (Li and Bou-Zeid, 2014). Thus, the estimation of T2 heavily depends on the values of the simulated TSK and T_level1. We have modified the inappropriate description in the revised manuscript. Please see the sentences marked in red in lines 248-252 of the revised manuscript.

**Q1.13:** Fig.5 would greatly benefit from the addition of subplots with the actual 2m temperature, Tskin, and HFX 2D fields for the default Kzmin value in of the ACM2 scheme. Moreover, there is no definition of the exact period that the authors consider as daytime and night-time.

**A1.13:** Thanks for the advice. We added the subplots as Fig. 6 in the revised manuscript as follows.

[Figure]

Figure A9. Spatial distribution of the mean difference (ACM2 1.0 minus ACM2 0.01) in (a, b) the 2-m temperature (T2), (c, d) the surface skin temperature (TSK), and (e, f) the sensible heat flux (HFX) over the daytime and the nighttime. The actual values of (g, h) T2, (i, j) TSK, and (k, l) HFX simulated by ACM2 with the default Kzmin

value (i.e. ACM2 0.01) during the daytime and the nighttime are also displayed for reference.

Moreover, the exact definitions of the daytime and the nighttime in the present study are 8-17 LST and 18-7 LST, respectively. We now clearly stated this information in the revised manuscript. Please see lines 148-149.

**Q1.14:** The manuscript would benefit from a through editing check.
**A1.14:** Thanks. We have carefully revised our manuscript again and tried our best to correct inappropriate statements and grammatical mistakes in the manuscript. Please see the contents marked in red throughout the revised manuscript.

**References:**

Pleim, J. E., 2007a: A combined local and nonlocal closure model for the atmospheric boundary layer. Part I: Model description and testing. J. Appl. Meteor. Climatol., 46 , 1383–1395.

Pleim, J. E., 2007b: A combined local and nonlocal closure model for the atmospheric boundary layer. Part II: Application and evaluation in a mesoscale meteorological model. J. Appl. Meteor. Climatol., 46 , 1396–1409.

Hu, X., Nielsen-Gammon, J. W., & Zhang, F., 2010. Evaluation of Three Planetary Boundary Layer Schemes in the WRF Model, Journal of Applied Meteorology and Climatology, 49(9), 1831-1844.

Nielsen-Gammon, J. W., Hu, X.-M., Zhang, F., and Pleim, J. E.: Evaluation of planetary boundary layer scheme sensitivities for the purpose of parameter estimation, Monthly Weather Review, 138, 3400–3417, 2010.

Li, D. and Bou-Zeid, E.: Quality and sensitivity of high-resolution numerical simulation of urban heat islands, Environmental Research 565 Letters, in press, 055001, 2014.

**Response to Referee #2**

The authors wish to thank Reviewer #2 for the comments, which greatly contribute to an improvement of the paper.

In the following, we address the issues raised by the reviewer:

**Q2.1:** This study only focuses on surface temperature (T2). How about water vapor and winds? You may improve temperature performance at the cost of worse performance for other variables. One cannot simply focus on one variable but ignore other variables during model calibration/optimization.

**A2.1:** The main objective of the present study is to estimate the influence caused by the change of Kzmin on the prediction of the near-surface temperature. Thus, we focused more on the temperature prediction in the manuscript. Actually, we have also performed comparisons of other meteorological parameters such as the wind speed with the observational data, which is presented below.

The values of statistical parameters for simulations of 2-m temperature ($^{\circ}$C), 10-m wind speed (m s$^{-1}$) and 2-m specific humidity (i.e. Q2) (g kg$^{-1}$) are listed in Tab. A1 of this rebuttal. It can be seen that the changes of the wind speed and the specific humidity caused by the altering of Kzmin are significantly smaller than that of the 2-m temperature. It means that the simulation accuracy of the wind speed and the specific humidity is not heavily influenced by the change of Kzmin. Thus, we paid more attention to the influence on the temperature prediction brought about by the change of Kzmin.

Table A1 Statistical parameters for simulations of 2-m temperature ($^{\circ}$C), 10-m wind speed (m s$^{-1}$) and 2-m specific humidity (i.e. Q2) (g kg$^{-1}$).

| Kzmin | T2 | | | | W10 | | | | Q2 | | | |
|---|---|---|---|---|---|---|---|---|---|---|---|---|
| | RMSE | IOA | R | MB | RMSE | IOA | R | MB | RMSE | IOA | R | MB |
| 0.01 | 2.79 | 0.84 | 0.94 | -2.49 | 3.21 | 0.26 | 0.64 | 2.51 | 0.21 | 0.85 | 0.73 | 0.02 |
| 0.2 | 2.32 | 0.88 | 0.93 | -2.00 | 3.27 | 0.26 | 0.62 | 2.60 | 0.21 | 0.84 | 0.72 | 0.01 |
| 0.5 | 2.01 | 0.90 | 0.92 | -1.58 | 3.31 | 0.25 | 0.63 | 2.70 | 0.21 | 0.85 | 0.73 | 0.01 |
| 0.8 | 1.88 | 0.91 | 0.90 | -1.27 | 3.38 | 0.25 | 0.62 | 2.77 | 0.20 | 0.85 | 0.74 | 0.01 |
| 1.0 | 1.82 | 0.91 | 0.89 | -1.08 | 3.39 | 0.25 | 0.62 | 2.79 | 0.20 | 0.86 | 0.75 | 0.00 |

The time series of the 10-m wind speed and the 2-m specific humidity (i.e. Q2) simulated in five scenarios using different constant Kzmin values are shown in Fig. A1 of this rebuttal. It can be seen that the increase of Kzmin exerts a small influence on the change of the wind speed and the specific humidity. Moreover, from Fig. A1(a), it was found that the increase of Kzmin induces a stronger 10-m wind speed. In contrast, the specific humidity Q2 was found lower under a higher Kzmin (see Fig. A1(b)). It is because that a larger Kzmin causes a stronger mixing. As a result, the

momentum is transported downwards from higher altitudes and the moisture is transported upwards from altitudes near the surface. Thus, a larger 10-m wind speed and a lower Q2 were obtained in the present study when a higher Kzmin is used.

[Figure]

Figure A1 Time series of 10-m wind speed (WS) and 2-m specific humidity (Q2) simulated by five scenarios with different Kzmin values.

Although the impact on 10-m wind speed and 2-m specific humidity caused by the change of Kzmin is relatively minor, based on the suggestions of the reviewer, we still added the related content about the validations of the simulated wind speed and the specific humidity in the revised supplementary material. Please see Sect. 4 of the supplementary material and the corresponding discussions.

**Q2.2:** Only data at one urban site are used in model evaluation in this study. The cold bias at this site may not be representative. Normally, regular soundings are used for PBL scheme calibration. Why don't use soundings?

**A2.2:** Thanks a lot for the suggestion of the reviewer. We added the content about the evaluation of 2-m temperature and 10-m wind speed at five observation stations in the revised manuscript in Section 3.1. These five observation stations include the IAP, CAS station and four AWS stations (Automatic Weather Stations). The information of the IAP station has been given in the original manuscript. Regarding to the four AWS stations, the No. 54433 AWS station is located in urban area of Beijing, similar to the IAP station, while the other three AWS stations (No. 54406, 54419, 54501) are located in rural or suburban areas of Beijing.

Table A2. Values of statistical parameters measuring the model performance in simulating the 2-m temperature (T2) and the 10-m wind speed (W10) at five observation stations.

| Station | T2 | | | | W10 | | | |
| --- | --- | --- | --- | --- | --- | --- | --- | --- |
| | RMSE | IOA | R | MB | RMSE | IOA | R | MB |
| IAP | 2.79 | 0.84 | 0.94 | -2.49 | 3.21 | 0.26 | 0.64 | 2.51 |
| 54406 | 2.84 | 0.88 | 0.83 | 1.06 | 3.08 | 0.44 | 0.50 | 2.21 |
| 54419 | 3.16 | 0.85 | 0.86 | 1.11 | 2.28 | 0.30 | 0.35 | 1.49 |
| 54433 | 2.17 | 0.92 | 0.91 | -1.38 | 2.40 | 0.62 | 0.65 | 1.26 |
| 54501 | 4.85 | 0.76 | 0.78 | 2.75 | 2.06 | 0.52 | 0.36 | 0.94 |

The values of statistical parameters measuring the model performance are listed in Tab. A2 of this rebuttal. From a global view, the model behavior in capturing the 2-m temperature is satisfying. The correlation coefficients between the simulated temperature and the observations at these five stations reside in a value range of 0.78-0.95. Moreover, the index of agreement (i.e. IOA) also possesses a value above 0.75 for all these five stations. It was also found that the model performs better at the two urban stations (IAP and No. 54433) than at the other three rural stations, denoted by a relatively smaller RMSE and a higher R (see Tab. A2).

In contrast to that, the model tends to predict a higher wind speed at all five stations (see RMSE of W10 in Tab. A2). The deviation between the simulations and the observational data is more obvious at the IAP station, as it possesses the largest RMSE of 3.21 m/s. Moreover, from the values of the correlation coefficient R, it was found that the simulated trend of the 10-m wind speed at two urban stations (IAP and No. 54433) is more consistent with the observations than that at the rural stations, as the correlation coefficient R at these two urban stations reaches a value above 0.6.

[Figure]

Figure A2. Diurnal mean time series of the 2-m temperature (T2) at five stations.

Figure A2 of this rebuttal also shows the diurnal change of the averaged T2 at these five stations as well as the observations. It can be seen that at these stations, the highest T2 appears at approximately 15:00 local standard time (LST), while the lowest T2 appears at around 8:00 LST. This is also the reason why we focused on these two time points in the present study. Moreover, it was found that at the three rural stations (No. 54406, 54419 and 54501), the simulated 2-m temperature at these stations is higher than the observational value in the nighttime, while it is lower than the observation during

the daytime when the temperature is high. As a result, the simulated diurnal variation of T2 is weaker than the observations. On the contrary, at two urban stations, the cold bias appears during the whole day.

With respect to the 10-m wind speed, Fig. A3 shows the spatial distribution of the time-averaged 10-m wind speed over the nighttime. It was found that a larger wind speed is estimated in mountain areas than in plain areas, leading to a stronger wind shear in mountain areas than that in plain areas.

[Figure]

Figure A3. The spatial distribution of the mean 10-m wind speed over the nighttime.

Evaluations of the 2-m temperature and the 10-m wind speed at these five observation stations are added in the revised manuscript as a section "3.1 Model Evaluations". Please see lines 211-237 in the revised manuscript.

Currently we did not use the data from regular soundings in the present study, because the vertical observational data from the tower has a relatively high time resolution of 10 minutes. In contrast, the time resolution of regular soundings is lower, i.e. 12 hours. Aside from that, the observation tower is located next to the surface observation stations, while the locations of regular soundings are more remoted. Thus, the vertical observational data obtained from the tower are more consistent with the surface data achieved from the surface stations, compared with those from regular soundings. Thus, we prefer the observational data from the tower. However, if the reviewer insists on adding the comparison with regular soundings, we are happy to perform this comparison if the submission deadline of the manuscript revision can be extended further. Thanks again for the valuable suggestion.

**Q2.3:** Cold bias seen at the urban site in this study may be due to model errors associated with urban scheme and anthropogenic heat, as well as errors in initial model states particularly soil moisture. So, this study may be attributing other model errors to PBL scheme.

**A2.3:** Thanks for the suggestion. The reviewer possibly thought that we were looking for a better PBL scheme in capturing the change of near-surface temperature over Beijing area in the present study. However, the focus of this study is to figure out the effect of using various Kzmin on the prediction of the temperature, instead of finding an improved PBL scheme. We are clearly aware that the deviation between the simulation results and the observational data can be caused by many other factors rather than the modification of the PBL scheme as the reviewer pointed out. Moreover, even we found the simulation results using Kzmin =1.0 closer to observations, we cannot guarantee that the scheme using Kzmin =1.0 is an improved scheme which is more realistic, because the better performance of the modified scheme can be caused by the offset of biases introduced by other factors. In order to avoid the misunderstandings that the biases all come from the modifications of the PBL scheme, we have revised all the related contents throughout the manuscript. Please see lines 108-109, 230-234, 435-436 and 494-495 of the revised manuscript.

**Minor comments:**

**Q2.4:** LN27-30 These are old PBL schemes, not including modern PBL schemes, for example scale-aware schemes and mass flux schemes

**A2.4:** Thanks a lot for the information. We have added the contents introducing some modern PBL schemes in the revised manuscript. Please see lines 43-54 in the revised manuscript.

**Q2.5:** LN128-130
Aerosol effect may impact temperature, which is not considered by the model simulation in this study. So, this study ascribes the potential bias of not considering aerosol effects to vertical mixing.

**A2.5:** This question is similar to the question **Q2.3** above. The reviewer is correct saying that the existence of aerosols is able to significantly affect the behavior of the model and thus the deviation between the simulation results and the observations. However, as we stated in the answer **A2.3** that the main objective of the present study is not to find a more appropriate PBL scheme to reduce the deviation. Instead, in the present study, we actually tried to assess the influence on the change of temperature caused by using different Kzmin values. In order to indicate this issue more clearly, we added the information about the treatment of aerosols in the revised manuscript. Please see lines 145-146.

**Q2.6:** LN166, what is the Landusef value used in the simulation

A2.6: The Landusef value as well as PURB used in this study are given in Fig. A4 of the rebuttal. We have also included these figures in the revised supplementary material (see Fig. S2 of the supplements). Moreover, the related description is also added in the revised manuscript. Please see line 186 of the revised manuscript.

[Figure]

Figure A4. The spatial distributions of the land-use fraction of the urban category and the calculated PURB.

**Q2.7:** LN255

I don't think Kzmin would affect daytime prediction. It must be the residual effects from the nighttime impact

**A2.7:** This question is actually similar to the question **Q1.11** raised by Reviewer #1. The reviewer is correct saying that the nighttime bias can accumulate so that it would impact the daytime simulation. However, in ACM2, the implementation of Kzmin can be simply represented as follows:

$$Kz'=Kz+Kzmin,$$

which means that Kzmin is added to Kz to constitute a total vertical turbulent diffusivity Kz'. As a result, even though in the daytime when Kz is relatively large, Kzmin is still able to affect Kz' in areas that the turbulent mixing is relatively weak. We have added this information as well as the explanations in lines 319-323 of the revised manuscript.

Based on the information given above, we can conclude that the difference in the simulated temperature during the daytime brought about by the change of Kzmin is caused by the combined effect of the large temperature difference during the nighttime and the different turbulent mixing intensity during the daytime. To clarify it, we performed another numerical experiment. In this experiment (named AC_night_0.01), Kzmin was set to 0.01 during the nighttime (same as ACM2_0.01),

but 1.0 during the daytime (same as ACM2_1.0). By doing that, the contributions to the difference of the temperature by these two processes can be assessed separately.

The time-averaged vertical profiles of the potential temperature at 8 LST and 15 LST are shown in Fig. A5. In Fig. A5(a), potential temperature profiles belonging to AC_night_0.01 and ACM2_0.01 were found close to each other, due to the same values of the nighttime Kzmin used in these two scenarios. In contrast to that, in the daytime (see Fig. A5(b)), AC_night_0.01 was found predicting a higher temperature than ACM2_0.01, which is caused by the increase of Kzmin during the daytime and the stronger turbulent mixing.

In contrast, AC_night_0.01 was also found giving a lower temperature than ACM2_1.0, although a same Kzmin (=1.0) is used during the daytime in these two scenarios. Thus, the difference between AC_night_0.01 and ACM2_1.0 denotes the residual effect caused by the bias of the temperature during the nighttime.

[Figure]

Figure A5. The vertical profiles of the potential temperature predicted by ACM2_0.01, ACM2_1.0 and AC_night_0.01 at (a) 8 LST and (b) 15 LST, averaged over the simulated days.

Thus, according to this numerical experiment, we confirm that there are two primary processes causing the temperature difference during the daytime. One is the change of Kzmin during the daytime. When Kzmin is increased, the vertical mixing in the boundary layer is enhanced, which causes a stronger transport of the air from the upper layer into the boundary layer, resulting in a warmer boundary layer during the daytime. The other process is the residual effect caused by the change of Kzmin in the nighttime. It is because that different Kzmin results in a large deviation in the nearsurface temperature during the nighttime. This deviation would maintain until the daytime comes so that the prediction of the daytime temperature would also be affected.

The corresponding results and the related discussions are added in the revised manuscript. Please see lines 327-355.

**Q2.8:** In Figure 6.
Actually K=0.01 gives better temperature variation/cooling rate during nighttime. This simulation just has some systematic bias, which may not be due to PBL schemes, but due to other model/inputs bias/errors, for example, the systematic bias from urban scheme and uncertainties in initial land properties especially soil moisture.
**A2.8:** This question is similar to the question **Q2.3**. The reviewer is correct saying that K=0.01 predicts a better trend of the temperature change during the nighttime. However, the magnitude of the simulated temperature obtained by assuming K=1.0 was found closer to observations in the present study, compared to those using other Kzmin values. However, the objective of the present study is actually to find out the influence of the change of Kzmin on the estimation of temperature, instead of figuring out an improved PBL scheme. We are clearly aware that the deviation between the simulation results and the observational data can be caused by other factors as the reviewer pointed out. Even we found the simulation results using Kzmin =1.0 closer to observations, we cannot guarantee that the scheme using Kzmin =1.0 is an improved scheme which is more realistic, because the better performance of the modified scheme can be caused by the offset of biases introduced by other factors. In order to clarify the objective of the present study more clearly, we have revised our manuscript and made related modifications in the revised version of the paper, to avoid possible misunderstandings. Please see lines 230-234 of the revised manuscript.

**Q2.9:** ACM2_CMAQ should be identical as K=1 since urban fraction is 1.
**A2.9:** Thanks. In completely non-urban areas, ACM2_CMAQ is identical to K=0.01, while in completely urban areas, it is identical to the case K=1.0. We have made the statement more clearly in the manuscript, please see lines 189-190 of the revised manuscript.

**Q2.10:** Fig. 7, there is only one profile. Is this a profile averaged over several days or just at a specific time of a day? Please be explicit.
**A2.10:** This a profile averaged over the time period under the investigation of the present study. In order to avoid the misunderstanding of the reviewer, we refined the captions of the related figures in the revised manuscript. Thanks for the comment.

---

## Author Response (AR2)

**Response**

The authors sincerely thank the Editor for the positive evaluation of the manuscript and the valuable suggestions, which remarkably improves our paper.

In the following, we address the issues raised by the Editor:

**Q1.1:** However, as it is not clear where all the measurement sites are located, would it be possible to get these marked in the smallest nest in Figure 1. The description of the sites and used measurements are now still missing from section 2.1. The stations and measurements should be described there and not in the results section.
**A1.1:** Thanks a lot for the suggestion. According to this comment, we marked the locations of the IAP station and the four automatic weather stations in Fig.1 (b) and (c) in the revised manuscript and in this rebuttal. We also added the related description of the sites and used measurements in Section 2.1. Please see lines 125-127, 136-140 in the revised manuscript.

[Figure]

Figure 1. Description of (a) the locations of three nested domains, (b) an enlarged drawing of the terrain belonging to the innermost domain (i.e. D03), and (c) the spatial distribution of the land-use categories within D03. Locations of the IAP station and the four automatic weather stations (No. 54433, 54406, 54419, 54501) are also marked in (b) and (c).

**Q1.2:** The new sensitivity simulations AC_night_0.01 and AC_urban_1 should also be listed in Table 2 and explained in section 2.2.2 rather than in the results section at L329-332 and L389-391, respectively.

**A1.2:** Thanks for the suggestion. We included the new sensitivity tests (i.e. AC_night_0.01 and AC_urban_1) in Table 2 in the revised manuscript. The related explanations about these two sensitivity tests were also added in Section 2.2.2. Please see lines 204-210 in the revised manuscript.

**Table 2.** Scenarios simulated in the present study, with different setup of Kzmin in the ACM2 scheme.

| Type | Kzmin ($m^2\ s^{-1}$) | Name |
|---|---|---|
| Constant | 0.01 | ACM2_0.01 |
| | 0.2 | ACM2_0.2 |
| | 0.5 | ACM2_0.5 |
| | 0.8 | ACM2_0.8 |
| | 1.0 | ACM2_1.0 |
| Function | 0.01~1.0 | ACM2_CMAQ |
| Sensitivity test | 1.0 (daytime), 0.01 (nighttime) | AC_night_0.01 |
| | 1.0 (urban), 0.01 (non-urban) | AC_urban_1.0 |

**Q1.3:** I would advise that the language of the manuscript is reviewed by a native speaker. There are multiple points with issues and here just to indicate a few of them:
  L110: "we described" should be in present mode
  L114-115: "…we first evaluated the performance of the PBL scheme (ACM2) with different Kzmin values in simulating meteorological parameters such as the temperature, by comparing with the observational data, and…" sentence is not well written. Should be simply "…we first evaluated the performance of the PBL scheme (ACM2) with different Kzmin values in simulating observed meteorological parameters, and…"
  L122: Should be "are as follows"
  L334-335: "..due to the same values of the nighttime Kzmin used in these two scenarios.." would be more fluent as "…due to the same night-time Kzmin values used in both scenarios…"
  L335: "In contrast to that…" can simply be "In contrast…"
  L366: "figure out" -> "see"
**A1.3:** Thanks for the advice. We asked Dr. Holger Grosshans from Physikalisch-Technische Bundesanstalt (PTB) to completely revise the manuscript for us. Many expressions were modified, and the corrections were marked in red throughout the revised manuscript. Furthermore, we will order the English language copy-editing service provided by Copernicus Publications, if the manuscript gets accepted.

**Q1.4:** What comes to the code and data availability, it is not recommended just to have link to the model code as in addition appropriate reference to the data should be added (see https://www2.mmm.ucar.edu/wrf/users/citing_wrf.html)
**A1.4:** We added the corresponding reference after the link, please see line 530. Thanks.